# Artificial Intelligence Applications to Personalized Dietary Recommendations: A Systematic Review

**DOI:** 10.3390/healthcare13121417

**Published:** 2025-06-13

**Authors:** Xi Wang, Zhiyuan Sun, Hong Xue, Ruopeng An

**Affiliations:** 1Brown School, Washington University in St. Louis, St. Louis, MO 63130, USA; 2College of Physical Education, Yangzhou University, Yangzhou 225127, China; dx120220089@stu.yzu.edu.cn; 3Department of Health Administration and Policy, George Mason University, Fairfax, VA 22030, USA; hxue4@gmu.edu; 4Silver School of Social Work, New York University, New York, NY 10003, USA; a4605@nyu.edu

**Keywords:** personalized dietary recommendations, artificial intelligence (AI), chronic conditions

## Abstract

**Background/Objectives:** Personalized dietary recommendations are essential for managing chronic conditions such as diabetes and irritable bowel syndrome (IBS). However, traditional approaches often fall short in accounting for individual metabolic variability. This systematic review evaluates the effectiveness of artificial intelligence (AI)-generated dietary interventions in improving clinical outcomes among adults. **Methods**: Following PRISMA guidelines, we searched six electronic databases for peer-reviewed studies published between 19 November 2015 to 6 September 2024. Eligible studies included adults aged 18 to 91 who received AI-generated dietary recommendations based on biomarkers such as blood glucose, gut microbiome composition, and self-reported data. Study designs included randomized controlled trials (RCTs), pre-post studies, and cross-sectional analyses. The GRADE system was used to assess the quality of evidence. **Results**: Eleven studies met inclusion criteria (five RCTs, five pre-post designs, one cross-sectional). Most AI methods used in the included studies were based on machine learning (ML), including conventional ML algorithms, deep learning (DL), and hybrid approaches integrating ML with IoT-based systems. Interventions led to improved glycemic control, metabolic health, and psychological well-being. Notable outcomes included a 39% reduction in IBS symptom severity and a 72.7% diabetes remission rate. Among nine studies with comparison groups, six reported statistically significant improvements in AI groups, two found comparable or better outcomes, and one found no difference. Mild side effects such as fatigue and constipation were observed. **Conclusions**: AI-generated dietary interventions show promise in surpassing traditional approaches by providing personalized, data-driven recommendations. Further research is needed to validate long-term effects, refine intervention protocols, and enhance user adherence in both clinical and public health settings.

## 1. Introduction

Suboptimal dietary behaviors and increasingly sedentary lifestyles have contributed to the mounting global burden of chronic diseases such as diabetes, cardiovascular disorders, and gastrointestinal conditions in recent decades [1]. Although standard dietary guidelines and one-on-one consultations with dietitians have served as mainstays in disease prevention and management, they often fail to account for substantial inter-individual differences in metabolism, gut microbiota composition, genetic factors, and psychosocial contexts [2]. This variability underscores an urgent need for personalized dietary strategies capable of efficiently responding to the individual’s unique needs.

Recent advancements in artificial intelligence (AI) have begun to transform how we generate and deliver tailored dietary advice. Unlike traditional approaches that rely on generalized guidance or the experiential insights of dietitians, AI-driven methodologies apply conventional machine learning (ML) algorithms and deep learning (DL) models to integrate, interpret, and predict complex biological and behavioral data [3]. By mapping out sophisticated interactions among biomarkers, gut microbiome profiles, and dietary components, these AI systems can produce personalized recommendations that align more precisely with an individual’s physiological, metabolic, and lifestyle requirements. Moreover, the rise of wearable sensors and the expansion of real-time data acquisition—such as continuous glucose monitoring—further empower AI-based nutritional planning to adapt and refine interventions rapidly [4].

Preliminary studies suggest that AI-driven dietary recommendation systems can yield improved health outcomes, particularly in glycemic control, gastrointestinal symptom relief, and metabolic risk reduction [5]. However, the clinical efficacy and broader implications of these interventions remain incompletely understood. Heterogeneity in study designs, variances in AI algorithms used, and limitations in sample size complicate the interpretation and synthesis of findings. Additionally, ethical and practical barriers, including data privacy, algorithm transparency, user adherence, and concerns about algorithmic bias, pose significant challenges to implementing AI-guided diets at scale [6]. These gaps highlight the importance of systematically evaluating the performance and safety of AI-derived nutritional plans as well as the factors that may influence user acceptance and long-term adherence.

Several reviews have explored the intersection of AI and personalized nutrition, highlighting the transformative potential of AI in improving dietary recommendations. For instance, Kirk et al. (2022) conducted a review focused on conventional ML algorithm applications in nutrition science, emphasizing their ability to predict health outcomes based on dietary patterns and metabolic data [7]. Similarly, a systematic review by Patil et al. (2024) investigated the role of AI in analyzing gut microbiota data for nutritional recommendations, highlighting significant progress but also the limitations in current validation methods [8]. However, these reviews often focus narrowly on specific AI applications, such as predictive modeling or gut microbiota analysis, without examining their integration into broader clinical and real-world contexts. To address these gaps, our review provides a clinically focused, methodologically rigorous synthesis of AI-generated personalized dietary interventions across diverse health contexts, including diabetes and IBS. Unlike previous reviews, we systematically assess both algorithmic approaches (e.g., conventional ML, DL, and IoT-based hybrids) and clinical outcomes, while also evaluating evidence quality using standardized appraisal tools. Furthermore, we highlight implementation and ethical considerations, offering a more holistic perspective to guide real-world applications and future research. Additionally, most existing reviews lack an emphasis on the ethical, logistical, and practical considerations critical to translating AI-driven dietary systems into scalable, user-friendly tools.

This systematic review uniquely synthesizes evidence from diverse study designs—RCTs, pre-post interventions, and observational studies—to evaluate the holistic impact of AI-driven dietary recommendations across clinical and non-clinical settings. By critically appraising the methodological quality, reported outcomes, and challenges in implementation, our review aims to fill a significant gap in the literature. Notably, 7 of the 11 included studies were published in 2023–2024, reflecting the rapid growth and recent interest in AI-powered personalized dietary interventions. Specifically, it explores not only the efficacy but also the ethical and operational factors necessary for real-world adoption, offering actionable insights for future innovation in precision nutrition.

## 2. Materials and Methods

### 2.1. Overview

This review followed the Preferred Reporting Items for Systematic Reviews and Meta-Analyses (PRISMA) guidelines [9]. This review was registered in the OSF database under the title “Generalized Systematic Review Registration” (https://doi.org/10.17605/OSF.IO/ZWEQ3, accessed on 19 April 2025).

### 2.2. Study Selection Criteria

Predefined inclusion and exclusion criteria were established and applied to all identified studies during the screening process. The inclusion criteria for study selection encompassed experimental studies (e.g., randomized controlled trials [RCTs], pre-post interventions) and observational studies (e.g., cross-sectional), focusing on modern AI approaches, including conventional machine learning (ML) algorithms and deep learning (DL). Eligible studies included participants aged 18 to 91, comprising both healthy individuals and adults with prediabetes, diabetes, functional constipation, irritable bowel syndrome (IBS), depression, anxiety, or those undergoing hemodialysis. Data types considered included personal input data from blood samples, stool samples, and self-reported data. The intervention of interest was AI-generated dietary plans, with outcomes assessed based on individual health status after following such diets, including objective physiological measurements and subjective psychological assessments. We specifically focused on studies in which AI tools directly generated dietary recommendations and evaluated their impact on physical and mental health, rather than those that merely used AI methods to assist in the recommendation process. Only original, empirical, peer-reviewed journal publications written in English were included. The search covered studies published from 19 November 2015 to 6 September 2024.

The exclusion criteria ruled out studies that did not involve human subjects or lacked an observational or experimental design. Studies employing rule-based (“hard-coded”) approaches instead of example-based ML or DL were excluded, as were those involving non-human subjects or not utilizing personal input data. Interventions recommending nutritional supplements or medical treatments were not considered, nor were diets not directly recommended by AI, including those prescribed by experts or dietitians, even if based on AI-generated data. Studies were excluded if their outcomes were not directly related to personal health status. Additionally, letters, editorials, study or review protocols, case reports, review articles, commentaries, and conference abstracts were not included. These exclusion criteria are visually summarized in Figure 1, which now includes a breakdown of the reasons for excluding full-text articles.

### 2.3. Search Strategies

We systematically searched six electronic databases: Cochrane, EBSCO (including MEDLINE, Global Health, and CINAHL Plus with Full Text), EMBASE, SCOPUS, Web of Science, and PubMed. The search period spanned from 19 November 2015 to 6 September 2024. These databases were selected to ensure broad and interdisciplinary coverage of relevant literature in clinical nutrition, public health, biomedical sciences, and health informatics. We did not include computer science-specific repositories (e.g., IEEE Xplore, ACM DL), as our inclusion criteria focused on studies that implemented AI-generated dietary interventions in clinical, patient-centered, or real-world public health settings rather than algorithm development or theoretical model simulation.

The search strategy was constructed around four conceptual blocks:AI-related terms (“artificial intelligence,” “AI,” “computational intelligence,” “machine intelligence,” “computer reasoning,” “machine learning,” “deep learning,” “neural network/s,” “reinforcement learning,”)Personalization-related terms (personal*, individual*, tailor*, customize*, customize*).Diet-related terms (diet*, nutrition*, “meal plan*”)Recommendation-related terms (recommend*, advice, plan*, generate*)

These conceptual blocks were combined using Boolean operators (AND, OR) for each database and adapted to database-specific syntax and controlled vocabulary. Specifically, synonyms within each block were joined using OR, and the four conceptual blocks were combined using AND to ensure logical precision. We required at least one AI-related term to appear in the title to enhance precision. For EBSCO databases, we restricted our search to academic journals within MEDLINE, Global Health, and CINAHL Plus with Full Text. Complete search strategies for each database are provided in Appendix A. The complete search strategy, including database-specific modifications and detailed search strings, is provided in Appendix A. After the initial search, two reviewers (X.W. and Z.S.) independently screened titles and abstracts of all retrieved articles according to predefined inclusion criteria. The inter-rater reliability was assessed using Cohen’s kappa coefficient (κ = 0.84). Both reviewers subsequently examined full texts of potentially relevant articles. Disagreements were resolved through discussion between the reviewers, with additional consultation from R.A. when consensus could not be reached.

### 2.4. Data Extraction and Synthesis

We collected the following information through a standardized data extraction form: author, publication year, and details relevant to the subjects studied. This included the study design, intervention design, sample size, age distribution, and the proportion of females (see Table 1). We also gathered information on AI-recommended diet, health status, statistical approach, and estimated effect of dietary recommendations on health status (see Table 2). No meta-analysis was feasible, given the substantial heterogeneity of the models, outcome measures, and applications. For data extraction, one reviewer extracted study characteristics, AI methods, outcomes, and quality assessments using a standardized template. A second reviewer verified all extracted data for accuracy and completeness.

### 2.5. Study Quality Assessment

The Grading of Recommendations, Assessment, Development, and Evaluations (GRADE) system was used to assess the quality of evidence in the selected studies. GRADE categorizes evidence into four levels: high, moderate, low, and very low. While randomized controlled trials (RCTs) begin with a high evidence rating, observational studies start at low quality due to potential confounding factors. During the evaluation, a study’s evidence level can be upgraded or downgraded based on five key factors: risk of bias, imprecision, inconsistency, indirectness, and publication bias.

We systematically applied the GRADE criteria to each included study based on its design, reporting clarity, and methodological rigor. For example, studies were downgraded when they had small sample sizes, lacked control groups, or failed to report key demographic or methodological details. Conversely, studies with well-documented randomization procedures and appropriate statistical analyses were retained at high or moderate quality levels.

The final quality ratings for all studies are summarized in Table 1 and were used to inform our interpretation of the strength of the evidence throughout the Section 3 and Section 4.

## 3. Results

### 3.1. Identification of Studies

Figure 1 illustrates the PRISMA flow diagram, outlining the structured literature search and selection procedure. A total of 1859 articles were identified through systematic database searches, comprising 333 from Cochrane, 279 from EBSCO, 24 from EMBASE, 351 from PubMed, 568 from Scopus, and 314 from Web of Science. Following duplicate removal, 1131 unique articles were screened by two independent reviewers (X.W. and Z.S.) at the title and abstract level, excluding 1105 articles. The most common reasons for exclusion at this stage included: (1) absence of AI-related methods or terminology in the title or abstract; (2) lack of dietary interventions or focus on unrelated topics (e.g., genomics, pharmacology, or purely behavioral interventions); (3) non-human studies (e.g., animal models or in silico simulations); (4) studies not involving personalized or individualized nutrition (e.g., population-level dietary guidelines); and (5) articles lacking original empirical data, such as editorials, commentaries, protocols, or reviews. These reasons have now been reflected in the updated PRISMA diagram to improve transparency in the screening process.

The remaining 26 articles underwent full-text evaluation against the predefined selection criteria. Subsequently, 15 articles were excluded for the following reasons: conference abstracts (*n* = 7), non-AI-generated dietary interventions (*n* = 4), AI-generated supplements or nutritional components rather than complete diets (*n* = 3), and ongoing experimental studies (*n* = 1). There are 11 articles included in our system review [10,11,12,13,14,15,16,17,18,19,20].

### 3.2. Study Characteristics

Table 1 presents the essential characteristics of the included studies in this systematic review, while Table 2 summarizes the outcome measures and quantitative impact of AI-generated dietary interventions. The included studies were published between 2015 and 2024, with the majority (*n* = 10) published within the past five years. They were conducted in various countries and regions, including Israel [11,12,13,17], Turkey [14,19], the USA [10,15], India [16], the UK [18], and China [20].

**Table 1 healthcare-13-01417-t001:** Essential Characteristics of the Studies Included in the Review.

Study ID	Author, Year	Country /Region	Study Design	Intervention Design	Sample Size	Age	Female (%)	Grade
1	Zeevi, 2015 [11]	Israel	RCT ^c^	Healthy individuals were randomly assigned to the prediction arm (*n* = 12) and the expert arm (*n* = 14). Every participant followed a personalized “good” diet and “bad” diet (categorized by AI ^e^ or experts) for a full week in total 2 weeks.	26	N/A ^h^	N/A	Moderate
2	Ben-Yacov, 2021 [12]	Israel	RCT	Adults with prediabetes were randomly assigned to follow a MED ^f^ diet (*n =* 112) or a PPT ^g^ diet (*n =* 113) for 6 months.	225 (200 completed 6-month intervention; 177 completed 12-month follow-up)	43–57	132 (58.7)	High
3	Rein, 2022 [13]	Israel	Pre-post	Short-term Crossover Intervention: 23 newly diagnosed T2D ^d^ patients received both diets, with the order randomized, each diet for 2 week, total 4 week. Long-term Intervention: 16 out of the original 23 participants continued PPT diet for 6 months.	23	45–62	12 (52)	Moderate
4	Arslan, 2022 [14]	Turkey	RCT	Patients with FC ^n^ were randomly assigned to follow a conventional treatments group or an AI diet group for 10 weeks, with 6 weeks for AI diet intervention.	45	21–42	40 (88.9)	High
5	Connell, 2023 [15]	USA	Pre-post	Four interventional trials (single-arm for IBS ^i^, depression, and anxiety; two-arm for T2D) to evaluate the efficacy of a VPNP ^j^, with the T2D trial lasting 8–9 months.	IBS-SSS ^k^ study, *n =* 105 (severe = 52, moderate = 41, mild = 12); Depression study, *n =* 410 (severe = 108, moderate = 129, mild = 173); Anxiety study, *n =* 490 (severe = 54, moderate = 124, mild = 312); T2D Study, *n =* 2912 (low adherence group = 1456, high adherence group = 1456)	31–66 (for the T2D trial)	T2D trial: 1808 (62)	Moderate
6	Joshi, 2023 [16]	India	RCT	Adult participants with diabetes were randomly assigned to a DT ^l^ group or SC ^m^ group for 1 year.	*n =* 319 (DT = 233, SC = 86)	18–70	N/A	High
7	Ben-Yacov, 2023 [17]	Israel	RCT	Adults with prediabetes were randomly assigned to follow a MED diet or a PPT diet for 6 months.	200	18–65	N/A	High
8	Bul, 2023 [18]	UK	Pre-post	Adult participants with prediabetes or diabetes or their carers use the AI-driven web-based nutrition platform for 8 weeks.	Pre: 73 Post: 23	27–79	Pre: 58 (79)	Moderate
9	Tunali, 2024 [19]	Turkey	Pre-post	IBS patients were randomly assigned to PD ^a^ and FODMAP ^b^ diets for 6 weeks supervised by a dietitian who was blinded to the diets.	121 (PD = 70, FODMAP = 51)	18–65	42 (60)	Moderate
10	Jin, 2024 [20]	China	Pre-post	In Phase 1, HD ^o^ patients receive standard dietary education regarding potassium management in HD. In Phase 2, they received GPT-based dietary guidance for 1 week.	88	48–73	N/A	Moderate
11	Buchan, 2024 [10]	USA	Cross-sectional	N/A	2420	18–91	1765 (72.9)	Low

Notes: ^a^ PD: artificial intelligence-assisted personalized diet; ^b^ FODMAP: low-fermentable oligosaccharides, disaccharides, monosaccharides, and polyols diet; ^c^ RCT: Randomized Controlled Trial; ^d^ T2D: type 2 diabetes; ^e^ AI: Artificial Intelligence; ^f^ MED: Mediterranean-style MED diet; ^g^ PPT: personalized postprandial glucose targeting diet; ^h^ N/A: not applicable; ^i^ IBS: irritable bowel syndrome; ^j^ VPNP: personalized precision nutrition program; ^k^ IBS-SSS: Irritable Bowel Syndrome—Symptom Severity Scale; ^l^ DT: Digital Twin; ^m^ SC: Standard Care; ^n^ FC: functional constipation; ^o^ HD: hemodialysis.

**Table 2 healthcare-13-01417-t002:** Measures and Estimated Impact of AI-recommended Diet.

Study ID	Author, Year	AI-Recommended Diet	Health Outcome Measures	Statistical Tests	Estimated Effect of AI-Recommended Diet on Health Outcomes
1	Zeevi, 2015 [11]	In the prediction arm, the “good” diet was designed using machine learning to predict individual postprandial glycemic responses (PPGR ^k^) based on each participant’s personal data, selecting meals from their prior dietary records that were expected to minimize PPGRs.	Postprandial Responses: PPGR Glucose fluctuations Max PPGR	Wilcoxon signed-rank test	Based on the Wilcoxon signed-rank test, across all participants, the “good” diet exhibited significantly lower PPGRs than the “bad” diet (*p* < 0.05); lower fluctuations in glucose levels across the CGM ^al^ connection week (*p* < 0.05), and a lower maximal PPGR (*p* < 0.05) in the “good” diet. In the prediction arm, the “good” diet is negatively associated with PPGR, M (good diet) vs. M (bad diet): 19 vs. 30 mg/dL.h, *p* < 0.001; In expert arm, “good” diet is negatively associated with PPGR, M (good diet) vs. M (bad diet): 18 vs. 39 mg/dL.h, *p* < 0.001. In the prediction arm, good diet is negatively associated with Glucose fluctuations, M (good diet) vs. M (bad diet): 0.12 vs. 0.14 σ/μ, *p* < 0.01; In the expert arm, good diet is negatively associated with PPGR, M (good diet) vs. M (bad diet): 0115 vs. 0.155 σ/μ, *p* < 0.01. In the prediction arm, good diet is negatively associated with Glucose fluctuations, M (good diet) vs. M (bad diet): 55 vs. 98 mg/dL.h, *p* < 0.01; In the expert arm, good diet is negatively associated with PPGR, M (good diet) vs. M (bad diet): 40 vs. 85 mg/dL.h, *p* < 0.01. The success of the prediction arm was comparable to that of the expert-based arm: In the predictor arm, 10 out of 12 participants showed significantly lower PPGRs (*p* < 0.05); in the expert arm, 8 out of 14 participants showed significantly lower PPGRs (*p* < 0.05).
2	Ben-Yacov, 2021 [12]	PPT diet (Dietary recommendations were personalized based on PPGR using a machine-learning algorithm that integrates clinical and microbiome data, selecting meals individually scored for each participant from a meal bank.)	Primary outcomes: Daily hours with glucose levels > 140 mg/dL. HbA1c ^j^ Secondary outcomes: Triglycerides HDL ^p^ cholesterol Total cholesterol-to-HDL cholesterol ratio FLI ^q^	Two-sample *t*-test, Mann-Whitney nonparametric test	Based on the two-sample *t*-test, the difference was significantly more significant in the PPT group than in the MED group after 6-month intervention: Daily hours with glucose levels > 140 mg/dL: M(SD) change: −0.3 (0.8) h/day for MED diet and −1.3 (1.5) h/day for PPT diet, with 95% CI between-group difference −1.29 to −0.66 h/day, *p* < 0.001. HbA1c: M(SD) change: −0.9 (2.1) mmol/mol for the MED diet and −1.7 (2.6) mmol/mol for the PPT diet, with 95% CI between-group difference −0.14 to −0.02% (1.5 to 0.2 mmol/mol), *p* = 0.007. Triglycerides: M(SD) change: −0.22 (0.51) mmol/L [−19 (45) mg/dL] for MED diet, and −0.43 (0.58) mmol/L [−38 (51) mg/dL] for PPT diet. 95% CI between-group difference −0.36 to −0.07 mmol/L [−31.51 to −6.11 mg/dL], *p* = 0.003. HDL cholesterol: M(SD) change: 0.02 (0.18) [0.8 (6.7) mg/dL] for MED diet and 0.09 (0.22) mmol/L [3.6 (8.5) mg/dL] in for PPT diet; 95% CI between-group difference 0.02–0.13 mmol/L [0.77–4.9 mg/dL] *p* = 0.003. Total cholesterol-to-HDL cholesterol ratio (Based on Mann-Whitney nonparametric test): M(SD) change: −0.29 (0.73) for MED diet, and −0.37 (0.71) for PPT diet; 95% CI between-group difference −0.3 to −0.00, *p* = 0.025. FLI: M(SD) change: 7.4 (3.7) for MED diet and 13.1 (14.3) for PPT diet; 95% CI between-group difference 8.34 to 1.41, *p* = 0.005. The significant between-group differences were maintained at the 12-month follow-up.
3	Rein, 2022 [13]	In the crossover phase, participants in PPT diet had limited (4–5) meal options, while in the long-term phase, they had hundreds of choices, selected based on a scoring system using their machine-learning-predicted PPGR	Crossover Intervention: Average PPGR Mean glucose Daily time of glucose levels > 140 mg/dL Blood fructosamine levels Long-term Intervention: HbA1c Fasting glucose Triglycerides	Two-sample *t*-test, linear mixed models, Wilcoxon signed-rank test	In the crossover intervention: PPT diet is negatively associated with Fructosamine change (95% CI [−32.41, −0.11]; *p* = 0.048) PPGR (95% CI [−0.04, −0.01]; *p* < 0.001) Blood glucose fluctuations (95% CI [−26.86, −12.79]; *p* < 0.001) The PPT diet led to significantly lower levels of CGM-based measures as compared to the MED diet: PPGR: MD between diets: −19.8 ± 16.3 mg/dL × h; *p* < 0.001 Blood glucose: MD between diets: −7.8 ± 5.5 mg/dL; *p* < 0.001 Daily time of glucose levels > 140 mg/d: MD between diets: −2.42 ± 1.7 h/day; *p* < 0.001 Blood fructosamine: MD between diets: − 16.4 ± 37 μmol/dL, *p* < 0.0001 In the long-term intervention: PPT diet is negatively associated with HbA1c change, M(SD): −0.39 (0.48) %, *p* < 0.001 FPG change, M(SD): −16.4 (24.2) mg/dL, *p* = 0.02 Fructosamine change, M(SD): −26.7 (22.5) μmol/dL, *p* < 0.001 Fasting insulin change, M(SD): −2.3 (4.0) MCU/mL, *p* = 0.04 Mean CGM glucose change, M(SD): −7.2(10.9) mg/dL, *p* = 0.02 Mean CGM glucose change, M(SD): −7.2(10.9) mg/dL, *p* = 0.02 Daily time with glucose levels change > 140 mg/dL, M(SD): −1.88 (2.89) h/day, *p* = 0.02 Based on the Mann-Whitney U test, the PPT diet is negatively associated with HOMA-IR change, M(SD): −5 (4.1), *p* < 0.001
4	Arslan, 2022 [14]	The AI-assisted (Enbiosis Biotechnology, Sariyer, Istanbul) diet was created by analyzing each patient’s gut microbiome, using machine learning to generate personalized nutritional recommendations focused on fiber-rich foods, and adjusting weekly based on patient feedback and progress.	CBMpW ^aj^, PAC-QoL ^ak^ scores: physical discomfort, psychosocial discomfort, worries and discomfort, satisfaction	Paired *t*-test	Based on the paired-sample *t*-test, AI diet is positively associated with CBMpW, M(SD): post: 4.3 (1.8) vs. pre: 1.7 (1.6), *p* < 0.001. Based on the paired *t*-test, AI diet is negatively associated with total PAC-QoL score, M(SD): post: 15.9 (16) vs. pre: 52.1 (16.9), *p* < 0.001. Customization of a diet based on individual microbiome tests provides better outcomes both clinically and socially in FC patients compare to conventional treatment group.
5	Connell, 2023 [15]	Precision food and personalized supplement recommendations computed by the VPNP ^l^ (based on each person’s stool or blood samples, Viome AI Recommendation Engine generates functional scores to determine final food and supplement recommendations based on machine learning.)	IBS: IBS-SSS Depression: PHQ9 ^m^ score Anxiety: GAD7 ^n^ score T2D ^o^ risk score (0 to 100)	N/A	Using VPNP is negatively associated with IBS in all kinds of categories. For the severe group, the post-mean score of IBS-SSS is 39% lower than pre, *p* = 0.0058. Using VPNP is negatively associated with depression in all kinds of categories. For the severe group, the post-mean score of PHQ9 is 31% lower than pre, *p* < 0.0001. Using VPNP is negatively associated with anxiety in all kinds of categories. For the severe group, the post-mean score of GAD7 is 31% lower than pre, *p* < 0.0001. Using VPNP is negatively associated with T2D risk score in the high adherence group, M(SD): post: 41.59 (23.53) vs. Pre: 71.84 (25.19), *p* < 0.0001. Besides, comparing the low adherence group to the high adherence group, Participants who adhered to precision recommendations reduced their T2D risk score more than participants who did not, *p* < 0.001.
6	Joshi, 2023 [16]	Digital twin (DT) technology is used to predict PPGRs to specific foods (machine learning algorithms and Internet of Things (IoT)-integrated systems) and provide real-time dietary recommendations via a mobile app.	Primary outcomes: HbA1c Reduction in diabetes medication use Secondary outcomes: Diabetes remission rates. Liver function test scores, including: NAFLD-LFS ^r^, NAFLD-NFS ^s^, FLI, FSI ^t^, FIB-4 ^u^, APRI ^v^; Visceral adiposity: MRI ^w^ Liver fat percentage: MRI-PDFF ^x^ Weight loss, BMI ^y^, WC ^z^, HOMA2-IR ^aa^ Liver enzymes: ALT ^ab^, AST ^ac^, GGT ^ad^; Glycemic variability measures: CV ^ae^, MAGE ^af^; Inflammatory markers: hs-CRP ^ag^, WBC ^ah^, ESR ^ai^ Ferritin levels	Paired *t*-test	For diabetes remission rates, at 1 year, T2D remission was achieved in 152 (72.7%) of 209 DT patients based on A1C compared with none in the SC group. Basing on paired *t*-test, using a novel nudge-touch DT-enabled personalized nutrition technology is negatively associated with many indexes: Weight, M(SD): post: 71.0 (13.5) vs. pre: 78.4 (14.5), *p* < 0.001 BMI, M(SD): post: 24.5 (4.0) vs. pre: 27.2 (4.4), *p* < 0.001 WC, M(SD): post: 88.1 (9.1) vs. pre:97.6 (11.2); *p* < 0.001 HbA1c, M(SD): post: 6.1 (0.7) vs. Pre: 9.0 (1.9), *p* < 0.001 HOMA2-IR, M(SD): post: 1.2 (0.6) vs. Pre: 1.9 (0.9), *p* < 0.001 ALT (U/L), M(SD): post: 30.3 (20.5) vs. Pre: 44.6 (32.5), *p* < 0.001 AST (U/L), M(SD): post: 23.4 (11.2) vs. Pre: 27.9 (14.8), *p* < 0.001 GGT (U/L), M(SD): post: 26.5 (25.3) vs. Pre: 30.9 (28.2), *p* < 0.001. NAFLD-NFS, M(SD): post: −3.0 (1.0) vs. Pre: −2.3 (1.1), *p* < 0.001 NAFLD-LFS, M(SD): post: −0.9 (1.7) vs. Pre: 1.2 (2.0), *p* < 0.001 FLI, M(SD): post: 34.1 (25.4) vs. Pre: 61.9 (24.5), *p* < 0.001 FSI, M(SD): post: 24.2 (22.7) vs. Pre: 56.1 (25.9), *p* < 0.001 APRI, M(SD): post: 0.3 (0.2) vs. Pre: 0.3 (0.3), *p* < 0.001 CV (%), M(SD): post: 18.4 (6.2) vs. Pre: 22.9 (6.4), *p* < 0.001 MAGE (mg/dL), M(SD): post: 25.3 (16.2) vs. Pre: 35.4 (17.4), *p* < 0.001 hs-CRP (mg/L), M(SD): post: 1.3 (1.7) vs. Pre: 2.9 (4.5), *p* < 0.001 WBC (cells/mm^3^), M(SD): post: 6595.6 (1751.8) vs. Pre: 7750.2 (1892.9), *p* < 0.001 ESR (mm/h), M(SD): post: 8.7 (7.8) vs. Pre: 13.4 (10.1), *p* < 0.001 Basing on paired *t*-test, using a novel nudge-touch DT-enabled personalized nutrition technology is positively associated with many indexes: FIB4, M(SD): post: 0.8 (0.4) vs. Pre: 0.7 (0.3), *p* < 0.001 Ferritin (ng/mL), M(SD): post: 93.8 (94.7) vs. Pre: 13.4 (10.1), *p* < 0.001 The study demonstrates that DT-enabled personalized nutrition significantly improves glycemic control, liver fat reduction, and metabolic dysfunction-associated fatty liver disease (MAFLD) markers compared to standard care (SC). No such significant change in SC group.
7	Ben-Yacov, 2023 [17]	PPT diet (Dietary recommendations were personalized based on PPGR using a machine-learning algorithm that integrates clinical and microbiome data, selecting meals individually scored for each participant from a meal bank) and PPT-adherence scores ranging from 0 to 100.	HbA1c HDL Triglycerides	Causal mediation analysis	Basing on the causal mediation analysis, PPT diet adherence is negatively associated with HbA1C_Blood (coefficient = −0.0015; *p* = 0.0048) and Triglycerides (coefficient = −0.0053; *p* = 0.0007); Basing on the causal mediation analysis, PPT diet adherence is positively associated with HDL (coefficient = 0.0019; *p* = 0.0014) The PPT diet group showed statistically significant improvements in multiple metabolic and lipid indicators, while the changes in the MED group were smaller or not significant.
8	Bul, 2023 [18]	The AI-driven nutrition platform uses deep learning techniques to analyze user data and context, generating personalized meal plans and recipe suggestions from personally viewed or saved recipes and popular trending recipes online.	General health status (scores 0–100): EuroQol Visual Analog Scale (EQ-VAS) Diabetes-related health indicators: Height, weight, waist circumference, HbA1c, systolic and diastolic blood pressure, HDL, total cholesterol levels	Wilcoxon matched pairs signed-rank test	Basing on the Wilcoxon matched pairs signed-rank test, using the platform is not statistically associated with general health status (*n =* 23, MD −1.7, 95% CI −9.0–6.0; *p* = 0.61; Cliff δ = −0.05). Basing on the Wilcoxon matched pairs signed-rank test, using the platform is negatively associated with weight (MD 4.5 kg/m2, 95% CI 1.0–12.0; *p* = 0.009; Cliff δ = 0.33) and waist size (*n =* 23, mean difference 3.9 cm, 95% CI 2.0–6.5; *p* = 0.008; Cliff δ = 0.48).
9	Tunali, 2024 [19]	Following the microbiome analysis, PD ^a^ and FODMAP ^b^ diets were administered. The PD was planned with the foods recommended by AI (based on machine learning models) according to the results of the microbiome analysis. Approximately 300 foods were scored between 0 and 10 for microbiome modulation.	IBS-SSS ^c^: IBS-C ^d^, IBS-D ^e^ and IBS-M ^f^ IBS-QOL ^g^: IBS-C, IBS-D and IBS-M Anxiety: HADS ^h^ Depression: HADS	Paired *t*-test, Wilcoxon signed-rank test	Basing on the paired *t*-test or Wilcoxon signed-rank tests, PD diet is negatively associated with IBS-SSS. In total, IBS-SSS, M(SD) ^i^: post: 210.64 (130.63) vs. pre: 314.42 (92.79), *p* < 0.001; For IBS-C, M(SD): post: 201.68(122.67) vs. pre: 327.91(97.74), *p* < 0.001; For IBS-D, M(SD): post: 221.13 (126.24) vs. pre: 306.06 (100.88), *p* = 0.010; For IBS-M, M(SD): post: 187.41 (148.24) vs. pre: 300.86 (80.01), *p* < 0.001. Basing on the paired *t*-test or Wilcoxon signed-rank test, PD diet is positively associated with IBS-QOL. In total, IBS-QOL, M(SD): post: 55.79 (21.85) vs. pre: 45.55 (22.06); For IBS-C, M(SD): post: 55.30 (22.90) vs. pre: 45.86 (22.25), *p* < 0.001; For IBS-D, M(SD): post: 58.41 (20.83) vs. pre: 45.82 (25.21), *p* < 0.01; For IBS-M, M(SD): post: 54.60 (21.85) vs. pre: 44.89 (20.35), *p* = 0.08. Basing on the paired *t*-test or Wilcoxon signed-rank tests, PD diet is negatively associated with Hospital Anxiety. M(SD): post: 8.15 (3.37) vs. Pre: 10.27 (4.22), *p* < 0.001. Basing on the paired *t*-test or Wilcoxon signed-rank test, PD diet is negatively associated with HADS. M(SD): post: 6.22 (4.10) vs. Pre: 7.57 (4.35). No significant difference between two groups in all the scores.
10	Jin, 2024 [20]	A custom GPT-based tool that analyzed uploaded food photographs, estimated potassium content, and provided real-time personalized dietary advice based on guidelines for hemodialysis patients.	Predialysis serum potassium levels Proportion of hyperkalemia: predialysis serum potassium > 5.0 mmol/L, Predialysis pH levels	Mixed-effects linear regression, Chi-square test	Basing on mixed-effects linear regression analysis, GPT diet is negatively associated with predialysis serum potassium levels, M(SD): post: 4.57 (0.76) mmol/L vs. pre: 4.84(0.94) mmol/L, *p* = 0.004. Basing on Chi-square test, GPT diet is negatively associated with Proportion of hyperkalemia, proportion: traditional diet: 39.8% vs. GPT diet: 25.0%, *p* = 0.036. Basing on mixed-effects linear regression analysis, GPT diet is not statistical significantly associated with predialysis pH levels, M(SD): post: 7.383 (0.040) vs. pre: 7.380 (0.035), *p* = 0.632. The GPT diet performs better than the standard diet in reducing serum potassium levels: Mean predialysis serum potassium level: M(SD): GPT: 4.57 (0.76) mmol/L vs. Standard: 4.84 (0.94) mmol/L, *p* = 0.004.
11	Buchan, 2024 [10]	A text-based virtual dietitian, Ina (based on machine learning algorithms), is designed to provide cancer patients with personalized nutritional support and guidance.	User experience: a 5-point Likert scale satisfaction survey Adherence: a 5-point Likert scales satisfaction survey Symptom management: a 5-point Likert severity scale survey QoL: a 7-point Likert scale survey	Descriptive statistics	93.6% of users were satisfied with the platform; 83.9% reported actively using advice to guide their diets; 87.7% reported the platform helped them manage symptoms; 81.4% felt the program improved their quality of life. 32.4% out of 785 users had a reduction in total number of symptoms; and 34.0% had a reduction in total symptom severity score. 61.8% of respondents experienced nutrition-related side effects: fatigue (42.8%), constipation (17.9%), dry mouth (15.4%), etc.

Notes: ^a^ PD: artificial intelligence-assisted personalized diet; ^b^ FODMAP: low-fermentable oligosaccharides, disaccharides, monosaccharides, and polyols diet; ^c^ IBS-SSS: Irritable Bowel Syndrome—Symptom Severity Scale ^d^ IBS-C: Irritable Bowel Syndrome with Constipation; ^e^ IBS-D: Irritable Bowel Syndrome with Diarrhea; ^f^ IBS-M: Irritable Bowel Syndrome with Mixed bowel habits; ^g^ IBS-QOL: Irritable Bowel Syndrome—Quality of Life; ^h^ HADS: Hospital Anxiety and Depression Scale; ^i^ M(SD): Mean (Difference standard deviation); ^j^ HbA1c: Hemoglobin A1c; ^k^ PPGR: average postprandial glucose responses; ^l^ VPNP: Viome AI Recommendation Engine; ^m^ PHQ9: Patient Health Questionnaire-9; ^n^ GAD7: Generalized Anxiety Disorder-7; ^o^ T2D: Type 2 diabetes; ^p^ HDL: high-density lipoprotein; ^q^ FLI: Fatty Liver Index; ^r^ NAFLD-LFS: Non-Alcoholic Fatty Liver Disease Liver Fat Score; ^s^ NAFLD-NFS: Non-Alcoholic Fatty Liver Disease Fibrosis Score; ^t^ FSI: Fatty Liver Index Score; ^u^ FIB-4: Fibrosis-4 Score, ^v^ APRI: AST to Platelet Ratio Index; ^w^ MRI: Magnetic Resonance Imaging-Proton; ^x^ MRI-PDFF: Magnetic Resonance Imaging-Proton Density Fat Fraction; ^y^ BMI: Body Mass Index; ^z^ WC: Waist circumference; ^aa^ HOMA2-IR:Homeostatic Model Assessment 2 of Insulin Resistance; ^ab^ ALT: Alanine Aminotransferase, ^ac^ AST: Aspartate Aminotransferase, ^ad^ GGT: Gamma-Glutamyl Transferase; ^ae^ CV: Coefficient of Variation; ^af^ MAGE: Mean Amplitude of Glycemic Excursions; ^ag^ hs-CRP: high-sensitivity C-Reactive Protein; ^ah^ WBC: White Blood Cell count; ^ai^ ESR: Erythrocyte Sedimentation Rate; ^aj^ CBMpW: Complete Bowel Movements per Week; ^ak^ PAC-QoL: Patient Assessment Constipation, Quality of Life; ^al^ CGM: Continuous Glucose Monitoring.

The intervention duration ranged from 2 to 52 weeks. Sample sizes ranged from 23 to 2420 participants. Specifically, 5 studies had sample sizes between 10 and 99 [10,12,13,17,19]; 1 study between 99 and 199 [19]; 3 studies between 200 and 299 [12,17,19]; 1 study between 300 and 499 [16]; 2 studies between 1000 and 2500 [10,15].

The participants were all adults, ranging from 18 to 91 years old. 1 targeted healthy adults [11], 2 focused on prediabetes adults [12,17], and 6 addressed patients with diabetes [16], type 2 diabetes (T2D) [13,15], functional constipation (FC) [14], irritable bowel syndrome(IBS) [15,19], and hemodialysis (HD) [20]. 1 focused on the prediabetes adults or diabetes patients [18]. The percentage of female participants ranged from 52% (12/23) to 88.9% (40/45) across the included studies.

The included studies employed diverse research methodologies. Five were randomized controlled trials (RCTs) [11,12,14,16,17,18], comprising three two-arm RCTs and two single-arm RCTs. Five studies utilized pre-post designs with varying approaches: two incorporated comparison or comparator groups [15,19], one combined a short-term cross-over trial [13], and two were conducted without control groups [18,20]. The remaining study employed a cross-sectional design [10].

### 3.3. AI-Driven Diet Recommendation System

Regarding AI-recommended diet systems, most studies employed machine learning algorithms (e.g., Viome AI Recommendation Engine) [10,11,12,13,14,15,16,17,19], while two studies used deep learning techniques (e.g., AI-driven nutrition platforms and ChatGPT 4.0-based systems) [18,20]. One study combined the Internet of Things (IoT)—integrated systems with machine learning algorithms to provide real-time dietary recommendations [16]. Table 2 summarizes the AI-based dietary recommendation measures employed in the included studies. The recommendation systems typically comprise two main components: (1) analyzing individual physical health indicators and (2) dietary recommendations based on these analyzed measurements. In this review, deep learning (DL) is categorized as a subfield of machine learning (ML), and we consistently use ML as an umbrella term where appropriate.

#### 3.3.1. Analyzing Individual Physical Health Indicators

Assessing individual physical conditions relies on three primary data sources: blood samples, stool samples, and self-reported data. Regarding blood-based measurements, Postprandial Glucose Response (PPGR) is the most frequently utilized indicator [11,12,13,16,17]. Some specialized indicators, such as predialysis serum potassium levels, were specifically employed for dialysis patients [10]. Regarding stool analysis, the primary focus was on examining individual gut microbiome profiles [14,19], including specialized microbiome analyses [19]. While most studies concentrated on a single data source, some investigations incorporated blood and stool sample analyses [15]. For self-reported data, participants provided health-related metrics such as height, weight, waist circumference, blood glucose levels, etc. [10,18].

#### 3.3.2. Dietary Recommendations

The dietary recommendation component encompasses both recommendation content and guidelines. Regarding content, some studies based their recommendations on subjects’ previous or current dietary patterns [11,18,20], while others selected diets from specially constructed meal banks designed for the recommendation system [12,13,17,19]. Specific dietary recommendations were generated through machine learning models trained on domain knowledge, including publications on microbial and human physiology, food science, and clinical trials [10,15]. Other studies utilized Internet of Things (IoT)-based recipes [18], while some studies did not specify their dietary recommendation sources [14,16].

Concerning recommendation guidelines, most studies employed AI-learned correlations between physical indicators and food items to predict and recommend dietary effects [10,11,12,13,14,15,16,17,18,19]. Some studies, however, based their recommendations on classical dietary guidelines, such as Jin (2024) [20], which utilized the Mayo Clinic’s Renal Diet Handbook and standard guidelines for chronic kidney disease (CKD) patients.

### 3.4. Measure of Health Status

The included studies evaluated health status using both objective physiological measurements and subjective psychological assessments. Seven of the 11 studies reviewed reported blood-based measurements [11,12,13,16,17,18,20], while three incorporated stool analyses [14,15,19].

#### 3.4.1. Physiological Measurements

Blood-based parameters encompassed several key categories of health indicators. Glycemic control was the most frequently assessed parameter, evaluated in six studies [11,12,13,16,17,18]. Multiple glycemic measures were employed, including Postprandial Glucose Response (PPGR) for short-term glucose fluctuations [11,13], Glycated Hemoglobin (HbA1c) for long-term glucose levels [12,16,17,18], and Coefficient of Variation (CV) for glycemic variability [16]. Cardiovascular health indicators were examined in four studies [12,13,17,18] through lipid profiles (triglycerides, HDL cholesterol) and blood pressure measurements (systolic and diastolic). Hepatic health evaluation, conducted in two studies [12,16], utilized the Fatty Liver Index (FLI) to quantify hepatic fat accumulation. Metabolic status assessment, performed in two studies [13,16], incorporated visceral adiposity measurements (MRI) and insulin resistance (HOMA2-IR). One study [16] examined inflammatory markers, including high-sensitivity C-reactive protein (hs-CRP) and White Blood Cell count (WBC), to assess systemic inflammation.

Stool-based assessments were conducted in three studies, employing specific metrics for gastrointestinal health evaluation. The IBS Symptom Severity Scale (IBS-SSS) was utilized in two studies [15,19] to assess irritable bowel syndrome intensity, while one study [14] quantified constipation severity through Complete Bowel Movements per Week (CBMpW).

#### 3.4.2. Psychological Assessments

Psychological status assessment was incorporated in four studies [14,15,18,19], encompassing various dimensions of mental health and quality of life. Anxiety evaluation employed the Generalized Anxiety Disorder-7 (GAD7) scale [15] and the Hospital Anxiety and Depression Scale (HADS) [19]. Depression screening was conducted using the Patient Health Questionnaire-9 (PHQ9) [15]. The impact of gastrointestinal conditions on daily living was assessed through the Patient Assessment of Constipation Quality of Life (PAC-QoL) [14]. Overall health-related quality of life was evaluated using the EQ Visual Analogue Scale (EQ-VAS) [18], which provides subjective health scores from 0–100, and the IBS Quality of Life (IBS-QOL) score [19], specifically designed for evaluating life quality in IBS-SSS subtypes.

### 3.5. Estimated Effect of AI-Recommended Diet

Compared to baseline results, AI-generated diets significantly improved glycemic control, metabolic health, and liver function and reduced gastrointestinal and psychological symptoms. Moreover, compared to other conventional dietary recommendations—such as the Mediterranean diet or dietitian-tailored plans—AI-generated diets demonstrated superior effectiveness.

#### 3.5.1. Enhanced Metabolic Health and Well-Being

AI-generated dietary interventions were associated with significant health benefits across glycemic, metabolic, liver, gastrointestinal, and psychological domains. Based on pre-post-study designs, all 11 included studies reported improvements in at least one health dimension. Table 3 maps the AI method, model type, and health domain targeted by each intervention, helping illustrate how different algorithmic strategies relate to specific outcome improvements.

**Table 3 healthcare-13-01417-t003:** Structured Comparison of AI Techniques and Outcomes.

Study ID	Study (Author, Year)	AI Technique	Target Outcome(s)	Key Findings
1.	Zeevi, 2015 [11]	ML ^a^	Postprandial glycemic control	↓ ^s^ PPGR ^b^‚ ↓ Glucose fluctuation
2.	Ben-Yacov, 2021 [12]	ML	Glycemic control, lipids, liver function	↓ HbA1c ^c^‚ ↓ Triglycerides‚ ↑ ^r^ HDL ^d^
3.	Rein, 2022 [13]	ML	Glycemic control, fructosamine	↓ PPGR‚ ↓ Glucose‚ ↓ Fructosamine
4.	Arslan, 2022 [14]	ML	Constipation, QoL ^e^	↑ CBMpW ^f^‚ ↓ PAC-QoL ^g^
5.	Connell, 2023 [15]	ML	IBS ^h^, Depression, Anxiety, T2D ^i^ risk	↓ IBS-SSS ^j^‚ ↓ PHQ-9 ^k^‚ ↓ GAD-7 ^l^‚ ↓ T2D score
6.	Joshi, 2023 [16]	ML + IoT ^m^ (Hybrid)	T2D, Liver, Inflammation	72.7% remission, ↓ HbA1c, ↓ FLI ^n^‚ ↓ hs-CRP ^o^
7.	Ben-Yacov, 2023 [17]	ML	HbA1c, lipids	↓ HbA1c‚ ↑ HDL‚ ↓ Triglycerides
8.	Bul, 2023 [18]	DL ^p^	Weight, Waist circumference	↓ Weight‚ ↓ Waist size
9.	Tunali, 2024 [19]	ML	IBS, Anxiety, Depression	↓ IBS-SSS, ↑ IBS-QOL‚ ↓ HADS ^q^
10.	Jin, 2024 [20]	DL	Potassium level (HD patients)	↓ Serum potassium‚ ↓ Hyperkalemia
11.	Buchan, 2024 [10]	ML	QoL, Symptom management	↑ QoL‚ ↓ Symptoms, side effects reported

Notes: ^a^ ML: Machine Learning; ^b^ PPGR: Low-fermentable Average Postprandial Glucose Responses; ^c^ HbA1c: Hemoglobin A1c; ^d^ HDL: high-density lipoprotein; ^e^ QoL: Quality of Life; ^f^ CBMpW: Complete Bowel Movements per Week; ^g^ PAC-QoL: Patient Assessment of Constipation—Quality of Life; Irritable Bowel Syndrome; ^h^ IBS: Irritable Bowel Syndrome; ^i^ T2D: Type 2 Diabetes; ^j^ IBS-SSS: Irritable Bowel Syndrome—Symptom Severity Scale; ^k^ PHQ-9: Patient Health Questionnaire-9; ^l^ GAD-7: Generalized Anxiety Disorder-7; ^m^ IoT: Internet of Things; ^n^ FLI: Fatty Liver Index; ^o^ hs-CRP: high-sensitivity C-Reactive Protein; ^p^ DL: Deep Learning; ^q^ HADS: Hospital Anxiety and Depression Scale; ^r^ ↑: indicates an increase; ^s^↓: indicates a decrease.

Glycemic control was a consistently reported outcome. Multiple studies demonstrated reductions in HbA1c, postprandial glucose response (PPGR), and glycemic variability [11,12,13,16,17]. For instance, a one-year digital twin intervention achieved a 72.7% diabetes remission rate among type 2 diabetes patients [16], while PPT-based diets outperformed conventional plans in lowering glucose excursions [12,13].

In terms of metabolic health, several studies reported decreased triglyceride levels and increased HDL cholesterol, indicating improved lipid profiles following AI-based dietary recommendations [12,13,16,17]. In addition, liver function improved, as evidenced by reductions in Fatty Liver Index (FLI), ALT, AST, and GGT levels in long-term interventions [12,16]. Although slight increases in FIB-4 were observed in one study [16], the values remained within the normal clinical range. Ferritin levels also increased from suboptimal to normal, suggesting improved iron status.

Gastrointestinal symptom management was reported in participants with irritable bowel syndrome (IBS) and functional constipation. One study observed a 39% reduction in IBS-SSS scores [15], while another reported significant gains in complete bowel movements and reductions in PAC-QoL scores [14]. Mental health outcomes also improved in several studies: participants experienced a 31% reduction in depression (PHQ-9) and anxiety (GAD-7) scores [15,19] and improved IBS-related quality of life metrics [19].

Significant anthropometric improvements—including weight loss, BMI reduction, and smaller waist circumference—were found in interventions using real-time or mobile-based AI systems [15,16,18]. User experience and adherence were also favorable: in a large-scale implementation study, 93.6% of users reported satisfaction with the AI platform, 87.7% reported improved symptom management and higher adherence was associated with more pronounced health benefits [10,17].

While generally safe, one study noted mild nutrition-related side effects, including fatigue (42.8%), constipation (17.9%), and dry mouth (15.4%) [10], highlighting the need for continued user-centered refinement.

#### 3.5.2. AI Diets Surpass Traditional Plans

Generally, AI-generated diets outperformed traditional dietary recommendations, such as Mediterranean or dietitian-tailored plans. Among nine studies with control or comparison groups, six reported that AI-generated diets had statistically significantly better clinical outcomes, two showed comparable or superior results, and one found no significant difference. Compared to the Mediterranean (MED) diet, Ben-Yacov et al. (2021) [12] found that the AI-recommend diet (PPT diet) led to more pronounced improvements in pre-post health indicators. Rein (2022) [13] further demonstrated that the PPT diet resulted in significantly lower CGM-based measures than the MED diet. Additionally, Ben-Yacov et al. (2023) [17] reported that participants on the PPT diet experienced statistically significant enhancements in various metabolic and lipid parameters, while changes in the MED group were minimal or insignificant.

Compared to conventional dietary plans, several studies have demonstrated the superiority of AI-recommended diets in various clinical scenarios. Arslan (2022) [14] reported that an AI-recommended diet provided both better clinical outcomes and improved social functioning in FC patients compared to a conventional treatment group. According to Joshi (2023) [16], at one-year follow-up, 152 out of 209 patients (72.7%) with T2D achieved remission based on A1C criteria using an AI-recommended diet. In contrast, none of the patients in the standard care group attained remission. Tunali (2024) [19] found that an AI-recommended diet effectively improved IBS symptoms, showing no significant differences from a FODMAP diet across all assessed scores. Finally, Jin (2024) [20] demonstrated that a GPT-based diet outperformed a standard diet in reducing serum potassium levels.

Moreover, Zeevi (2015) [11] found that an AI-recommended diet produced outcomes comparable to those achieved through expert-based guidance. Additionally, Connell (2023) [15] reported that participants who closely followed the precision AI-recommended diet experienced a more substantial reduction in T2D risk scores than those with lower adherence.

### 3.6. Study Quality Assessment

We assessed the quality of the studies included in the review using the GRADE framework (Table 1). Four studies were rated as “high” quality [12,14,16,17], six were rated as “moderate” [11,13,15,18,19,20], and one was rated as “low” [10]. The primary reason for the “low” rating was the cross-sectional or qualitative design, which is inherently more susceptible to confounding bias. Studies rated as “moderate” generally employed pre-post designs, which provide more robust causal inferences than cross-sectional or qualitative approaches yet still introduce potential confounding due to the absence of randomization. Notably, among the “moderate” studies, one was a randomized controlled trial (RCT); however, it lacked certain key information (e.g., the sample’s age range and percentage of female participants), which limited its quality rating. The studies receiving a “high” rating were RCTs that had been carefully designed and thus achieved stronger methodological rigor.

## 4. Discussion

### 4.1. Overview of AI-Driven Dietary Interventions

The findings of this systematic review suggest that AI-generated dietary recommendations have demonstrated potential for improving nutrition-related outcomes by incorporating ML and DL methods, aggregating large-scale dietary databases, and leveraging real-time health monitoring data [11,12,13,16,17,20]. At the same time, the reviewed literature indicates that these interventions require careful consideration of methodological and ethical aspects [10,11,12,16,18,19,20]. Although the included studies highlight possible benefits—such as enhanced glycemic control [11,16], improved adherence [10,17], and beneficial alterations in gut microbiota [12,14,17,19]—most also point to constraints in generalizability [11,13,14,15,16,19,20], sample sizes [11,13,14,17,18,20], and underlying algorithms [10,11,12,13,16,17,18,19].

### 4.2. Generalizability and Algorithmic Transparency

Many studies included in this review had limited sample sizes consisting of specific demographic groups, which restricts the ability to generalize findings to broader populations. This is particularly concerning given the significant dietary variations across different cultural, ethnic, and socioeconomic groups. The absence of diverse study populations in existing studies limits the scalability and applicability of AI-based dietary recommendations to heterogeneous real-world settings. Furthermore, algorithmic transparency remains a major concern in these studies. Several studies failed to provide detailed descriptions of the models used, including feature selection, training datasets, validation methods, etc. This lack of transparency hinders independent validation and replication, making it challenging to compare results across studies and establish best practices for AI-driven nutrition interventions. These limitations, combined with recurrent concerns about data quality and equity, underscore the importance of balanced perspectives when evaluating the role of AI in dietary interventions.

### 4.3. Cultural Relevance and Model Bias

The interest in AI-based personalized nutrition aligns with ongoing endeavors to address dietary variability across populations with differing metabolic profiles, cultural backgrounds, and health states. Several studies reported positive associations between AI-based diets and improved health indicators, including glycemic metrics and markers of liver function [11,12,13,16,17,20]. However, there was also evidence of incomplete or small-scale validation, which raises questions about the reliability and replicability of these findings [17,18]. In particular, the algorithms employed in many trials were rarely reported in detail, limiting insights into their assumptions, data processing, and performance metrics [10,11,12,13,16,17,18,19]. The insufficient details around algorithmic design complicate the task of replicating or comparing results across different studies, thus restricting consensus-building and best-practice guidelines.

Moreover, the reliance on self-reported dietary data introduces potential inaccuracies, and the underrepresentation of various demographic groups and dietary patterns in training datasets poses challenges to ensuring cultural suitability and fairness in AI-based recommendations [10,11,16,19]. Many AI algorithms have been trained using datasets that predominantly reflect Western dietary patterns, limiting their effectiveness in populations with distinct nutritional traditions and food availability. This lack of diversity in training data can lead to biased recommendations that may fail to meet the nutritional needs and food preferences of underrepresented populations. These observations align with findings in the broader literature, where model interpretability and population inclusivity have been consistently identified as critical factors for the successful implementation of AI solutions. Addressing these challenges requires a concerted effort to develop culturally adaptive AI models that incorporate diverse dietary data and account for varying food preferences, health conditions, and socioeconomic constraints.

### 4.4. Feasibility and Long-Term Sustainability

The studies included in this review also have implications for clinical practice and public health. Where AI platforms were tested in controlled trials or pre-post comparisons, they sometimes outperformed conventional diets, suggesting that machine learning approaches are capable of addressing individualized nutritional needs [12,17,20]. Nevertheless, the extent to which these gains can be sustained over longer periods remains unclear, because few of the identified studies tracked health outcomes beyond one year [16]. Additionally, difficulties in user adherence and the novelty of AI-driven interfaces could affect the real-world feasibility of these interventions [12,18]. Similar challenges have been noted in previous work on digital health tools, where user engagement tends to decline over time in the absence of adequate support, behavioral reinforcement, and integration into clinical workflows [19]. Taken collectively, these observations suggest that AI-based tools may offer advantages in tailoring diets and analyzing complex data, but that their durability and acceptance likely depend on structured, long-term strategies to maintain motivation and trust.

### 4.5. Limitations of the Review

The studies included in this review and the review process itself also exhibit limitations. While the search strategy encompassed multiple electronic databases, the focus on English-language, peer-reviewed publications may have excluded relevant studies published in other languages or in gray literature sources. The diversity of study designs and varying outcome measures precluded a meta-analytic synthesis, restricting the ability to quantify effect sizes across different contexts. In addition, none of the studies offered comprehensive details on cost-effectiveness or implementation logistics, which remain pivotal considerations for deploying AI-based dietary interventions at scale [10,15]. These constraints are consistent with broader discussions in the digital health and AI fields, where robust, large-scale evaluation is necessary before innovative technologies are adopted as standard practices [21].

### 4.6. Critical Gaps and Future Directions

Despite promising findings, the included studies share several limitations that restrict the strength and applicability of the evidence. First, most studies featured small sample sizes and lacked demographic diversity [11,13,14,17,18,20], which limits the generalizability of their findings to broader or underrepresented populations. The absence of representative dietary data—especially from non-Western settings—impairs the cultural relevance and fairness of AI-generated recommendations. For example, traditional Asian diets rich in fermented foods or plant-based condiments may interact with gut microbiota in ways not represented in current models, leading to reduced accuracy or cultural misalignment in AI-generated recommendations.

Second, many interventions were short-term (ranging from two weeks to six months), with few assessing long-term adherence, sustainability, or potential rebound effects [16,18]. Given the chronic nature of conditions such as diabetes and IBS, longer-term studies are essential for understanding the durability of AI-driven dietary changes.

Third, algorithmic transparency remains a critical concern. Several studies did not clearly describe their model architecture, feature selection processes, training datasets, or validation procedures [10,11,12,13,16,17,18,19]. Without such information, it is difficult to ensure ethical use, evaluate comparative effectiveness, or enable reproducibility.

Fourth, a number of studies relied heavily on self-reported outcomes such as dietary intake, symptom severity, or mood, which introduces potential recall and reporting bias [10,11,16,19]. In addition, adverse events were minimally documented in most studies, despite reports of nutrition-related side effects such as fatigue and constipation in at least one trial [10]. This lack of adverse event tracking limits our ability to assess the safety and clinical viability of these interventions over time.

Fifth, few studies assessed cost-effectiveness, equity, or integration into healthcare systems [10,12,15,16]. These gaps make it difficult to evaluate the scalability and real-world applicability of AI-powered dietary interventions.

Additionally, future studies should consider the ethical handling of sensitive dietary and biometric data, particularly as AI models increasingly rely on continuous data collection through wearable and mobile devices. Ensuring data privacy and informed consent is essential for building trust and avoiding misuse. Moreover, the integration of multimodal data sources—including microbiome profiles, biomarkers, behavioral patterns, and contextual factors—has the potential to improve personalization but requires careful harmonization and methodological rigor.

Moreover, the lack of reporting on intervention failures—such as nutrient imbalances or reduced adherence due to poor personalization—limits our ability to assess the full range of AI system performance. This omission may reflect a publication bias, as studies with null or negative findings are less likely to be published. Our review did not include grey literature or preprints, which may have led to the underrepresentation of unsuccessful interventions. Future systematic reviews should expand their scope to include non-peer-reviewed sources and unpublished trials to mitigate this bias and provide a more balanced evaluation of real-world effectiveness.

Notably, few of the included studies grounded their intervention designs in established behavioral theories or precision nutrition frameworks. While personalized feedback is a hallmark of AI interventions, its alignment with models of habit formation, such as reinforcement schedules or self-efficacy pathways, was rarely examined. Additionally, socioecological factors such as cultural norms, economic access, or family food environments were largely absent from algorithmic design. Future research should integrate behavioral science and contextual variables to enhance the interpretability and real-world relevance of AI-based dietary guidance.

Finally, while the majority of included studies reported statistically significant improvements in clinical indicators, not all outcomes demonstrated clear or substantial benefits from AI-generated diets. For instance, in Ben-Yacov et al. (2021) [12], the total cholesterol-to-HDL cholesterol ratio showed only a marginal between-group difference. These findings highlight the importance of evaluating AI efficacy on a per-outcome basis, recognizing that superiority is often domain-specific rather than universal.

### 4.7. When Does AI Work Best?

While the overall evidence supports the promise of AI-generated dietary interventions, their effectiveness appears to be domain- and context-specific. Based on our synthesis, we identify several key conditions under which AI-driven personalization is more likely to yield meaningful clinical benefits:

Rich Individual-Level Data Inputs: Interventions that incorporated continuous glucose monitoring, microbiome sequencing, or detailed food logging (e.g., Korem et al., 2017 [11]; Ben-Yacov et al., 2021 [12]) demonstrated more robust personalization and superior metabolic outcomes. In contrast, systems relying solely on baseline questionnaires or retrospective intake data often delivered less pronounced improvements.

Feedback Frequency and User Engagement: Studies offering regular, adaptive feedback through digital interfaces (e.g., smartphone apps or web portals) were associated with higher adherence and more sustained changes in dietary behavior [15,16,18]. This suggests that real-time engagement mechanisms enhance the behavioral impact of AI-generated advice.

Health Domain Specificity: AI systems performed better in managing conditions with well-defined and trackable biomarkers (e.g., blood glucose in diabetes) compared to areas like IBS or mood-related symptoms, where outcomes were more subjective and difficult to quantify [14,17,19].

Population Characteristics and Cultural Fit: Results were generally more favorable when interventions were aligned with local dietary customs and health literacy levels. For instance, higher success rates were observed in Israeli and U.S. populations using digital health tools. However, generalizability is limited due to the lack of trials in more diverse sociocultural settings, including regions with high fermented food intake, plant-based diets, or different glycemic norms.

Transparent and Explainable Algorithms: Studies with clear documentation of model inputs, personalization logic, and validation procedures allowed for better reproducibility and clinical trust [10,15,18]. Opaque AI systems with limited explainability undermined interpretability and potential adoption in healthcare settings.

This synthesis offers a practical framework for understanding under what conditions AI is most likely to improve dietary outcomes—and where its limitations may emerge. Future trials should align AI model design with these factors to enhance both clinical relevance and implementation feasibility.

## 5. Conclusions

This systematic review evaluates the application of AI-generated dietary interventions across diverse clinical and public health contexts. Most included studies utilized machine learning approaches—including conventional algorithms and deep learning—to personalize dietary recommendations for chronic conditions such as diabetes and irritable bowel syndrome (IBS). Evidence suggests that AI-driven interventions can outperform traditional dietary approaches in improving specific metabolic outcomes and tailoring meal plans based on individual inputs such as gut microbiota, glycemic response, and lifestyle data.

Despite these promising results, the reviewed studies revealed notable limitations, including small and demographically narrow samples, limited geographic and cultural diversity, short intervention durations, and insufficient reporting on adverse effects and ethical safeguards. Furthermore, few studies grounded their interventions in behavior change theory or assessed real-world scalability.

Future research should prioritize methodological transparency, equity in data representation, and integration with healthcare systems. Incorporating long-term follow-up, rigorous safety assessments, and socioecological frameworks will be essential to ensure the effectiveness, ethical integrity, and generalizability of AI-generated dietary interventions.

## Figures and Tables

**Figure 1 healthcare-13-01417-f001:**
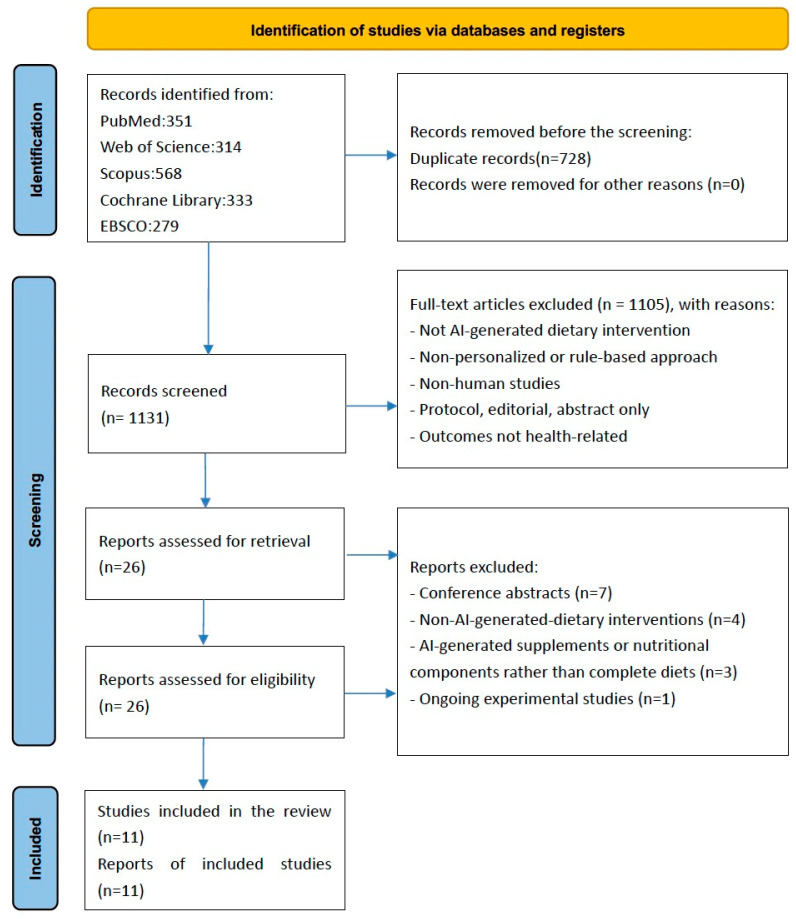
PRISMA (Preferred Reporting Items for Systematic Reviews and Meta-Analyses) flow diagram illustrating the study selection process.

## Data Availability

The data extracted and summarized in this systematic review (e.g., study characteristics and outcome measures) are available upon reasonable request from the corresponding author. No analytical code was used.

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
