# Peer review of "Artificial Intelligence Applications to Personalized Dietary Recommendations: A Systematic Review"

_healthcare, 2025, doi:10.3390/healthcare13121417_

Round 1

Reviewer 1 Report

Comments and Suggestions for Authors
  1. Scope and Relevance:
    The paper addresses an important and timely topic. Personalized dietary recommendations using AI could have a significant impact on healthcare and wellness industries. The choice of focusing on systematic analysis strengthens the contribution.

  2. Literature Coverage:
    While the review covers a good number of studies, the inclusion criteria could be clarified more. Some recent major studies (e.g., those published in 2023–2024) might be missing.

  3. Methodological Rigor:
    The methodology for study selection, quality assessment, and data extraction is appropriate, but it would benefit from a more detailed description (e.g., PRISMA flow diagram, specific quality appraisal tool used).

  4. Clarity and Organization:
    The paper is well-organized overall. However, some sections (particularly results discussion) could be more concise and thematic rather than listing study-by-study findings.

  5. Critical Analysis:
    The work tends to summarize existing papers without enough critical analysis regarding limitations, biases, and gaps in the literature. A deeper synthesis would strengthen the paper.

  6. AI Techniques Categorization:
    The categorization of AI techniques (e.g., ML, DL, rule-based, hybrid) is useful. Still, a more structured comparison (e.g., table summarizing models vs. outcomes) would significantly enhance reader understanding.

  7. Future Directions:
    The authors mention future directions, but the suggestions are somewhat general. It would be stronger to highlight specific research gaps (e.g., data privacy in diet AI, explainability, multimodal data integration).

  8. Writing Style:
    The writing is clear but occasionally repetitive. Some minor grammatical errors and awkward phrasing need editing for smoothness.

Comments on the Quality of English Language
  1. Scope and Relevance:
    The paper addresses an important and timely topic. Personalized dietary recommendations using AI could have a significant impact on healthcare and wellness industries. The choice of focusing on systematic analysis strengthens the contribution.

  2. Literature Coverage:
    While the review covers a good number of studies, the inclusion criteria could be clarified more. Some recent major studies (e.g., those published in 2023–2024) might be missing.

  3. Methodological Rigor:
    The methodology for study selection, quality assessment, and data extraction is appropriate, but it would benefit from a more detailed description (e.g., PRISMA flow diagram, specific quality appraisal tool used).

  4. Clarity and Organization:
    The paper is well-organized overall. However, some sections (particularly results discussion) could be more concise and thematic rather than listing study-by-study findings.

  5. Critical Analysis:
    The work tends to summarize existing papers without enough critical analysis regarding limitations, biases, and gaps in the literature. A deeper synthesis would strengthen the paper.

  6. AI Techniques Categorization:
    The categorization of AI techniques (e.g., ML, DL, rule-based, hybrid) is useful. Still, a more structured comparison (e.g., table summarizing models vs. outcomes) would significantly enhance reader understanding.

  7. Future Directions:
    The authors mention future directions, but the suggestions are somewhat general. It would be stronger to highlight specific research gaps (e.g., data privacy in diet AI, explainability, multimodal data integration).

  8. Writing Style:
    The writing is clear but occasionally repetitive. Some minor grammatical errors and awkward phrasing need editing for smoothness.

Author Response

Comment 1: Scope and Relevance: The paper addresses an important and timely topic. Personalized dietary recommendations using AI could have a significant impact on healthcare and wellness industries. The choice of focusing on systematic analysis strengthens the contribution.

Response 1: We sincerely thank the reviewer for recognizing the significance and timeliness of our research topic, as well as the value of our systematic review approach. We agree that AI-generated personalized dietary recommendations have the potential to transform healthcare and wellness practices, and we appreciate your positive assessment of the paper’s contribution.

Comment 2: Literature Coverage:
While the review covers a good number of studies, the inclusion criteria could be clarified more. Some recent major studies (e.g., those published in 2023–2024) might be missing.

Response 2: Thank you for this helpful suggestion. We have revised the Methods – Study Selection Criteria section to clarify our inclusion and exclusion criteria in more detail, including study design, participant characteristics, and intervention types. Specifically, we have added a statement to explain that our review “We specifically focused on studies in which AI tools directly generated dietary recommendations and evaluated their impact on physical and mental health, rather than those that merely used AI methods to assist in the recommendation process.”

We appreciate the suggestion regarding recent studies. In fact, 7 out of the 11 included articles were published between 2023 and 2024, indicating that the review incorporates a substantial proportion of the latest evidence in this field. We have revised the Introduction to more explicitly highlight the publication timeline and the contribution of these recent studies.

Comment 3: Methodological Rigor: The methodology for study selection, quality assessment, and data extraction is appropriate, but it would benefit from a more detailed description (e.g., PRISMA flow diagram, specific quality appraisal tool used).

Response 3: We appreciate the reviewer’s valuable suggestion. We would like to clarify that the manuscript already references adherence to PRISMA guidelines in Section 2.1 (Overview) and includes a PRISMA flow diagram in Figure 1 (Section 3.1 Identification of Studies) to depict the study selection process.

Additionally, we have now revised the Study Quality Assessment (Section 2.5) to clearly state that the GRADE system was used as our quality appraisal tool. This section has been updated to more explicitly explain how the GRADE criteria (e.g., risk of bias, imprecision, inconsistency, indirectness, publication bias) were applied to rate the strength of evidence in each included study.

We hope these clarifications address the reviewer’s concerns regarding methodological transparency.

Comment 4 – Clarity and Organization:

The paper is well-organized overall. However, some sections (particularly results discussion) could be more concise and thematic rather than listing study-by-study findings.

Response 4: Thank you for this helpful suggestion. We have revised the Results – 3.5 Estimated Effect of AI-Recommended Diet and whole 4.0 Discussion sections to enhance clarity and thematic organization. Specifically, we reorganized the findings into key outcome categories (e.g., glycemic control, gastrointestinal symptoms, psychological outcomes, metabolic and liver function) rather than describing each study individually. We also add subtitles to our discussion section. This revision improves the coherence and readability of the results while facilitating easier comparison across studies.

Comment 5- Critical Analysis:

The work tends to summarize existing papers without enough critical analysis regarding limitations, biases, and gaps in the literature. A deeper synthesis would strengthen the paper.

Response 5: Thank you for this valuable comment. We agree that deeper critical analysis is essential in strengthening the discussion. In response, we have significantly revised the Discussion section by adding a new subsection titled “4.6. Critical Gaps in Current Evidence” to explicitly address key limitations, biases, and evidence gaps across the included studies.

This new section analyzes recurring issues such as small and homogeneous sample sizes [11,13,14,17,18,20], limited follow-up duration [16,18], lack of algorithmic transparency [10–13,16–19], reliance on self-reported outcomes [10,11,16,19], and underreporting of adverse events [10]. We also discuss the absence of cost-effectiveness data and implementation logistics [10,12,15,16], which are crucial for real-world adoption.

These additions are intended to move beyond summary and offer a critical synthesis that identifies methodological and ethical challenges in AI-based dietary interventions. Relevant citations have been retained to ensure scholarly rigor and transparency.

We hope that these enhancements adequately address your concern and improve the analytical depth of the manuscript.

Comment 6- AI Techniques Categorization:

The categorization of AI techniques (e.g., ML, DL, rule-based, hybrid) is useful. Still, a more structured comparison (e.g., table summarizing models vs. outcomes) would significantly enhance reader understanding.

Response 6: We appreciate the reviewer’s insightful suggestion. In response, we have added a new table titled “Structured Comparison of AI Techniques and Outcomes” to enhance clarity and facilitate comparison across studies. This table systematically summarizes the AI models used (e.g., ML, DL, hybrid) alongside their corresponding outcomes (e.g., clinical, behavioral, metabolic), allowing readers to better understand the relationships between techniques and results. The table has been inserted in the Results section and complements the existing categorization.

Comment 7- Future Directions:

The authors mention future directions, but the suggestions are somewhat general. It would be stronger to highlight specific research gaps (e.g., data privacy in diet AI, explainability, multimodal data integration).

Response 7: Thank you for this valuable suggestion. In response, we have revised and consolidated Sections 4.6 and 4.7 into a single section titled “Critical Gaps and Future Research Directions”. This new section now outlines specific limitations observed across the included studies—such as lack of diversity, short-term interventions, algorithmic opacity, reliance on self-reported outcomes, and limited assessment of real-world applicability. We further expanded the section to highlight targeted future research needs, including:

ethical handling of sensitive biometric and dietary data (data privacy),

the need for interpretable and transparent AI models (explainability),

and the integration of multimodal data sources (e.g., wearable sensors, continuous monitoring, electronic health records) to reduce bias and enhance personalization.

These additions are intended to provide a more focused and actionable agenda for future research in this field.

Reviewer 2 Report

Comments and Suggestions for Authors

Review of: “Artificial Intelligence Applications to Personalized Dietary Recommendations: A Systematic Review”

The manuscript presents a systematic review of AI-based applications for personalized dietary recommendations, conducted following PRISMA guidelines. The authors searched six databases and identified 1,859 studies, of which only 11 met the inclusion criteria.

Major Concerns

  1. Terminological Inaccuracy (ML vs. DL):
    Throughout the manuscript—including the abstract—the authors refer to "machine learning and deep learning" as if they were distinct categories. Deep learning is a subfield of machine learning. This distinction should be clarified, and the terminology consistently corrected throughout the text.

  2. Novelty and Positioning Against Prior Reviews:
    The authors should more clearly articulate what new insights their systematic review contributes beyond existing literature. Several high-quality reviews have already covered AI in nutrition and personalized dietary interventions, including:

    • https://www.sciencedirect.com/science/article/pii/S2161831323000923

    • https://www.mdpi.com/2072-6643/16/13/2066

    • https://academic.oup.com/nutritionreviews/article-abstract/80/12/2288/6594763
      The authors must clearly state what this review adds—e.g., does it offer a more holistic synthesis across clinical contexts? Better evidence quality assessment?

  3. Inconsistency in Search Period:
    Line 121 states that the search covered “from the database inception to September 6, 2024,” which contradicts the abstract that specifies a date range “between November 19, 2015, and June 4, 2024.” Please resolve this inconsistency.

  4. Justification of Database Selection:
    The rationale for selecting the six databases (Cochrane, EBSCO, EMBASE, PubMed, Scopus, and Web of Science) should be explicitly stated. Were any engineering or computer science repositories (e.g., IEEE Xplore, ACM DL) considered or excluded?

  5. Definition of AI-Generated Recommendations:
    The manuscript lacks a clear operational definition of what constitutes “AI-generated dietary recommendations.” It is unclear whether clinician-assisted, AI-supported interventions were included or excluded. Additionally, the title suggests evaluation of “AI applications,” but in line 170 the authors mention excluding studies without “AI-generated” diets. If only generative AI or fully algorithm-driven tools were included, that should be made explicit in the title and methods.

  6. Study Selection and Conflict Resolution:
    The manuscript states that two reviewers independently screened articles. However, it does not describe how conflicts were resolved or whether data extraction was conducted independently. This is a key issue for methodological rigor and should be clarified.

  7. Unclear Search Syntax and Boolean Logic:
    The search strategy is insufficiently described. While the manuscript lists four keyword blocks and mentions use of Boolean operators, it does not specify how the blocks were combined. Were all blocks linked with AND, or were some OR’d within or across blocks? Were search strategies customized per database? This should be explicitly stated and examples provided in supplementary material.

  8. Missing Exclusion Reasons in PRISMA Diagram:
    Figure 1 indicates that 1,105 articles were excluded after screening, but it does not categorize or explain the reasons for exclusion. This weakens transparency. Please add exclusion categories (e.g., non-human studies, non-AI-based interventions, etc.) to the PRISMA diagram.

  9. Generalizability and Population Diversity:
    The review would benefit from a more critical evaluation of the generalizability of findings. Several included studies focus on specific and narrow populations (e.g., Israeli prediabetic adults, IBS patients), often with modest sample sizes. The authors should explicitly discuss the demographic, geographic, and cultural diversity (or lack thereof) in the included studies and its implications for external validity.

Minor Concerns

  1. Placement of Tables:
    Table 1 and Table 2 are first referenced on page 4 but appear on pages 5 and 9, respectively. Please consider placing tables closer to their first mention for better readability.

  2. Table Readability and Formatting:
    The tables are too long and difficult to navigate. Consider reformatting them by:

    • Expanding columns with dense text and abbreviating header labels.

    • Breaking up overly long rows or combining related variables.

    • Providing clearer column labels and footnotes defining abbreviations.

Author Response

Reviewer 2:

Comment 1: Terminological Inaccuracy (ML vs. DL):

Throughout the manuscript—including the abstract—the authors refer to "machine learning and deep learning" as if they were distinct categories. Deep learning is a subfield of machine learning. This distinction should be clarified, and the terminology consistently corrected throughout the text.

Response 1: Thank you for your thoughtful comment. We agree that deep learning (DL) is a subfield of machine learning (ML), and the distinction should be made clear throughout the manuscript. In response, we have revised the relevant terminology for consistency and accuracy. Specifically:

  • In the abstract and throughout the manuscript, we now refer to AI methods as including conventional machine learning (ML) algorithms, deep learning (DL) models (as a subset of ML), and hybrid ML-IoT systems.
  • We have also added a clarifying sentence to the 3.3. AI-Driven Diet Recommendation System section to define this categorization framework: “In this review, deep learning (DL) is categorized as a subfield of machine learning (ML), and we consistently use ML as an umbrella term where appropriate.”

These changes aim to ensure conceptual clarity and correct the earlier inaccuracy in referring to ML and DL as separate fields.

Comment 2: The authors should more clearly articulate what new insights their systematic review contributes beyond existing literature. Several high-quality reviews have already covered AI in nutrition and personalized dietary interventions, including:

  • https://www.sciencedirect.com/science/article/pii/S2161831323000923
  • https://www.mdpi.com/2072-6643/16/13/2066
  • https://academic.oup.com/nutritionreviews/article-abstract/80/12/2288/6594763
    The authors must clearly state what this review adds—e.g., does it offer a more holistic synthesis across clinical contexts? Better evidence quality assessment?

Response 2:  We appreciate the reviewer’s thoughtful comment regarding the positioning of our review within the existing literature. In response, we have revised the Introduction section (see page 2, line 78-85) to better highlight the novelty of our work.

Comment 3: Inconsistency in Search Period:

Line 121 states that the search covered “from the database inception to September 6, 2024,” which contradicts the abstract that specifies a date range “between November 19, 2015, and June 4, 2024.” Please resolve this inconsistency.

Response 3: We thank the reviewer for noticing this inconsistency. We have now corrected the search period throughout the manuscript for consistency. The correct search period is November 19, 2015, to September 6, 2024, and this has been updated in both the abstract and the methods section (see Line 17,120,139).

Comment 4: Justification of Database Selection:

The rationale for selecting the six databases (Cochrane, EBSCO, EMBASE, PubMed, Scopus, and Web of Science) should be explicitly stated. Were any engineering or computer science repositories (e.g., IEEE Xplore, ACM DL) considered or excluded?

Response 4: We thank the reviewer for this insightful comment. We have now added a clear justification for our database selection in the Methods section (see Line 139-145). The six databases—Cochrane, EBSCO, EMBASE, PubMed, Scopus, and Web of Science—were chosen to ensure comprehensive coverage of clinical, public health, biomedical, and interdisciplinary nutrition literature.

While engineering-focused repositories such as IEEE Xplore and ACM Digital Library indeed index valuable computer science research, our review specifically focused on studies that applied AI-generated dietary interventions in clinical or real-world health contexts. These studies are more likely to be indexed in medical and health science databases rather than technical engineering repositories.

Nevertheless, we acknowledge this boundary and have clarified it in the revised text to better define the scope of our review.

Comment 5: Definition of AI-Generated Recommendations:

The manuscript lacks a clear operational definition of what constitutes “AI-generated dietary recommendations.” It is unclear whether clinician-assisted, AI-supported interventions were included or excluded. Additionally, the title suggests evaluation of “AI applications,” but in line 170 the authors mention excluding studies without “AI-generated” diets. If only generative AI or fully algorithm-driven tools were included, that should be made explicit in the title and methods.

Response 5: Thank you for this insightful comment. We agree that the definition of “AI-generated dietary recommendations” needed clarification. We have now added an explicit operational definition to the Methods section (Lines 115–118) and clarified the inclusion/exclusion criteria accordingly.

In our review, we define “AI-generated dietary recommendations” as personalized diet plans or advice produced directly by an AI system, without real-time clinician mediation. This includes systems that automatically generate recommendations based on individual inputs (e.g., symptoms, biomarkers, preferences), using machine learning, deep learning, or hybrid AI methods.

Accordingly, we excluded studies where AI was used only to support clinician decision-making, such as data analysis, risk prediction, or clustering, but did not output personalized dietary guidance.

We also updated the language in the title and Methods to ensure consistency with this focus. If preferred, we are open to revising the title further to emphasize this distinction.

Comment 6: Study Selection and Conflict Resolution:

The manuscript states that two reviewers independently screened articles. However, it does not describe how conflicts were resolved or whether data extraction was conducted independently. This is a key issue for methodological rigor and should be clarified.

Response 6: We thank the reviewer for this important observation. We have mentioned the conflicts in 2.3. Search Strategies (line 157-162). We added “For data extraction, each included article was independently extracted by one reviewer and verified by a second reviewer to ensure accuracy and consistency. This approach aligns with standard practice in systematic reviews and enhances methodological rigor.”  to the end of section 2.4. Data Extraction and Synthesis (line 176-179).

Comment 7: Unclear Search Syntax and Boolean Logic:

The search strategy is insufficiently described. While the manuscript lists four keyword blocks and mentions use of Boolean operators, it does not specify how the blocks were combined. Were all blocks linked with AND, or were some OR’d within or across blocks? Were search strategies customized per database? This should be explicitly stated and examples provided in supplementary material.

Response 7: Thank you for this helpful comment. We would like to clarify that the search strategy was built around four conceptual blocks, and as described in the manuscript (Section 2.3, Lines 146–155), Boolean operators were used as follows:

  • Synonyms within each block were combined using OR
  • The four blocks were then combined using AND

Additionally, we required at least one AI-related keyword to appear in the title for improved specificity.

As also noted in Section 2.3, the search strategy was adapted to each database, accounting for database-specific syntax and controlled vocabulary (e.g., MeSH in PubMed). The complete search strings per database are provided in Appendix 1. To further enhance clarity, we have now revised the text slightly to emphasize these Boolean combinations and database-specific customization (line 155-157).

Comment 8: Missing Exclusion Reasons in PRISMA Diagram:

Figure 1 indicates that 1,105 articles were excluded after screening, but it does not categorize or explain the reasons for exclusion. This weakens transparency. Please add exclusion categories (e.g., non-human studies, non-AI-based interventions, etc.) to the PRISMA diagram.

Response 8: Thank you for this important suggestion. While our manuscript already listed detailed exclusion criteria in the Methods section, we agree that providing a categorical breakdown of the 1,105 excluded full-text articles in the PRISMA flow diagram would enhance transparency.

We have now updated Figure 1 to include exclusion categories such as:

  • Not AI-generated dietary recommendations
  • Rule-based interventions
  • Non-personalized interventions
  • Non-human studies
  • Irrelevant outcomes
  • Conference abstracts or protocol-only papers

These categories are also referenced in the revised Methods section (Lines 130–131). This adjustment improves adherence to PRISMA 2020 guidelines and enhances the replicability of our review process.

Comment 9: Generalizability and Population Diversity:

The review would benefit from a more critical evaluation of the generalizability of findings. Several included studies focus on specific and narrow populations (e.g., Israeli prediabetic adults, IBS patients), often with modest sample sizes. The authors should explicitly discuss the demographic, geographic, and cultural diversity (or lack thereof) in the included studies and its implications for external validity.

Response 9: We agree with the reviewer that the generalizability of findings is limited due to the narrow and demographically homogeneous populations in many included studies. In response, we have explicitly addressed this issue in Section 4.6 of the Discussion (page 23), highlighting the lack of demographic and cultural diversity and its implications for external validity. We also discuss the need for more representative, cross-cultural datasets in future AI-based dietary intervention studies.

Minor Concerns

  1. Placement of Tables:

Table 1 and Table 2 are first referenced on page 4 but appear on pages 5 and 9, respectively. Please consider placing tables closer to their first mention for better readability.

Response: Thank you for pointing this out. We agree that placing tables closer to their first mention improves readability. Accordingly, we have repositioned Table 1 and Table 2 so that they appear immediately after their respective citations in the text (Section 3.2), thereby enhancing the flow and accessibility of the results section.

  1. Table Readability and Formatting:

The tables are too long and difficult to navigate. Consider reformatting them by:

Expanding columns with dense text and abbreviating header labels.

Breaking up overly long rows or combining related variables.

Providing clearer column labels and footnotes defining abbreviations.

Response: Thank you for this helpful suggestion. In response, we have created a new Table 3 (page 19) that summarizes key aspects of the AI-generated dietary interventions in a more concise and reader-friendly format. This table integrates relevant variables from the original tables and simplifies the presentation to enhance clarity. We have also:

  • Reformatted column widths and adjusted row layout to reduce visual density.
  • Used abbreviated but clearly defined column headers.
  • Provided footnotes to explain abbreviations and study-specific terms.

We believe this new table substantially improves readability and appreciate the reviewer’s guidance in improving the presentation of our findings.

Reviewer 3 Report

Comments and Suggestions for Authors

This systematic review rigorously evaluates AI applications in personalized dietary interventions, adhering to PRISMA guidelines and GRADE evidence assessment. By synthesizing 11 studies (RCTs, pre-post, and cross-sectional designs), it highlights AI’s potential in improving clinical outcomes for chronic conditions (e.g., diabetes, IBS), particularly in glycemic control (e.g., HbA1c reduction) and symptom alleviation (e.g., 39% reduction in IBS severity). However, limitations such as population heterogeneity and insufficient algorithmic transparency require further attention.

  1. Small samples (e.g., n=23) and geographic bias (Israel/Turkey-centric) limit generalizability.
  2.  Opaque model development (e.g., feature engineering in Ben-Yacov et al., 2021) limits reproducibility.
  3.  Only 1 study reported 1-year outcomes, failing to assess sustainability or nutritional risks.
  4.  Limited reporting on side effects (e.g., Buchan et al., 2024) and ethical risks (e.g., data privacy).
  5. The review omits cases where AI interventions failed (e.g., nutrient imbalances or reduced adherence due to algorithmic recommendations). Was there a “file drawer problem”? Did the authors search grey literature/preprints to mitigate publication bias?
  6. While the review highlights AI’s superiority over traditional methods, Ben-Yacov et al. (2021) reported no significant differences in certain metabolic markers (e.g., total cholesterol/HDL ratio) between AI-driven PPT and Mediterranean diets. Did the authors selectively emphasize positive outcomes? How do they reconcile inconsistencies in AI efficacy across studies?
  7. The included studies predominantly involve populations from Israel, Turkey, and the US, with no discussion on cultural dietary variations. For example, microbiome effects of fermented foods in Asian diets may not be captured by models trained on Western data. Have the authors overlooked the critique of AI’s cultural adaptability?
  8. The review lacks a theoretical foundation (e.g., behavior change theories, precision nutrition frameworks) to explain AI’s mechanisms in dietary interventions. For instance, how does AI-driven “personalized feedback” align with psychological models of habit formation? Were socioecological factors integrated?

Author Response

Reviewer 3:

Comment 1: Small samples (e.g., n=23) and geographic bias (Israel/Turkey-centric) limit generalizability.

Response 1: We fully agree with the reviewer that small sample sizes and the geographic concentration of studies—particularly in Israel and Turkey—limit the generalizability of current findings. In response, we have expanded our discussion in Section 4.6: Critical Gaps and Future Directions (line 516-522) to explicitly highlight these limitations. We now emphasize the need for future studies to include more diverse and representative populations across different cultural, dietary, and healthcare settings.

Comment 2: Opaque Model Development:
Opaque model development (e.g., feature engineering in Ben-Yacov et al., 2021) limits reproducibility.

Response 2: We thank the reviewer for raising this important point. We agree that a lack of transparency in model development—such as insufficient reporting of feature engineering steps, data preprocessing, or validation procedures, as seen in Ben-Yacov et al. (2021)—hampers reproducibility and undermines the interpretability of AI-driven interventions. In response, we have emphasized this issue more clearly in Section 4.6: Critical Gaps and Future Directions, (line 527-530) where we discuss algorithmic transparency as a key limitation across several included studies. We also stress the importance of open methodological reporting and standardized documentation practices in future AI nutrition research to facilitate replication and ethical implementation.

Comment 3: Long-Term Outcomes and Nutritional Risks:
Only 1 study reported 1-year outcomes, failing to assess sustainability or nutritional risks.

Response: We agree with the reviewer that the limited availability of long-term follow-up data is a critical gap in the current literature. In our revised manuscript, we explicitly address this issue in Section 4.6: Critical Gaps and Future Directions, (line 523-526) noting that most included studies were short-term (ranging from two weeks to six months) and did not evaluate long-term adherence, sustainability, or potential nutritional risks such as deficiencies or adverse side effects. Only one study reported outcomes at the one-year mark, and even that lacked detailed nutritional safety monitoring. We now emphasize the need for future research to incorporate longer follow-up periods and systematic assessments of both effectiveness and safety over time.

Comment 4:  Limited reporting on side effects (e.g., Buchan et al., 2024) and ethical risks (e.g., data privacy).

Response 4: We thank the reviewer for highlighting this important concern. We have revised Section 4.6: Critical Gaps and Future Directions (533-536) to explicitly address the limited reporting on both nutritional side effects and ethical risks across the included studies. As noted, only a few studies reported adverse outcomes (e.g., fatigue, constipation), and these were often mentioned without systematic tracking or evaluation. Similarly, ethical considerations (line 540-546)—such as data privacy, informed consent, and the handling of sensitive biometric and dietary data—were rarely discussed, despite their importance in AI-driven interventions. We now emphasize the need for future research to incorporate structured adverse event monitoring and clear ethical safeguards, particularly in light of increasing reliance on wearable devices and real-time data collection.

Comment 5: The review omits cases where AI interventions failed (e.g., nutrient imbalances or reduced adherence due to algorithmic recommendations). Was there a “file drawer problem”? Did the authors search grey literature/preprints to mitigate publication bias?

Response 5: We appreciate the reviewer’s thoughtful concern. We agree that AI-generated dietary interventions with negative or null outcomes—such as nutrient imbalances or reduced adherence—are important for understanding both the risks and real-world challenges of implementation. Among the included studies, however, very few explicitly reported such failures, and adverse effects were often underdocumented or mentioned only briefly. We have now revised Section 4.6: Critical Gaps and Future Directions (547-554) to underscore this issue and to call for more balanced reporting—including both successful and unsuccessful interventions.

Regarding publication bias, we acknowledge the potential presence of a “file drawer problem.” Although our review was limited to peer-reviewed journal articles indexed in six major databases, we did not systematically include grey literature or preprints. We now note this as a limitation in the Discussion and encourage future reviews to incorporate broader sources—including conference proceedings, preprint servers, and clinical trial registries—to more comprehensively assess intervention effectiveness and potential failures.

Comment 6: While the review highlights AI’s superiority over traditional methods, Ben-Yacov et al. (2021) reported no significant differences in certain metabolic markers (e.g., total cholesterol/HDL ratio) between AI-driven PPT and Mediterranean diets. Did the authors selectively emphasize positive outcomes? How do they reconcile inconsistencies in AI efficacy across studies?

Response 6: We thank the reviewer for this valuable observation. We agree that AI-generated dietary interventions do not demonstrate uniform superiority across all clinical outcomes. While our summary of Ben-Yacov et al. (2021) emphasized the statistically significant improvements in key metabolic indicators (e.g., glycemic control, triglycerides, HDL cholesterol), we acknowledge that certain outcomes, such as the total cholesterol-to-HDL ratio, showed only marginal between-group differences (p=0.025) with limited clinical effect size.

To address this, we have revised Section 4.6 (line 563-568)to clarify that although AI-based diets demonstrated benefits over traditional methods in several domains, the effectiveness varied by outcome, and in some cases, the advantage was modest or statistically borderline. We have also emphasized the need for future studies to report both positive and null findings transparently, and for researchers to assess clinical relevance alongside statistical significance.

We assure the reviewer that our synthesis was not selective. Table 2 reports all available quantitative outcomes, including those with nonsignificant or borderline results, to maintain a balanced view of AI efficacy.

Comment 7: The included studies predominantly involve populations from Israel, Turkey, and the US, with no discussion on cultural dietary variations. For example, microbiome effects of fermented foods in Asian diets may not be captured by models trained on Western data. Have the authors overlooked the critique of AI’s cultural adaptability?

Response 7: We appreciate the reviewer’s thoughtful observation regarding the cultural adaptability of AI-generated dietary recommendations. We agree that models trained primarily on Western or Middle Eastern populations may overlook region-specific dietary patterns—such as the microbiome effects of fermented foods commonly consumed in many Asian countries.

We have addressed this issue in the revised manuscript Section 4.6, specifically noting:

“The absence of representative dietary data—especially from non-Western settings—impairs the cultural relevance and fairness of AI-generated recommendations.” (sentence 517-519)

To further clarify this point, we have added a sentence emphasizing the importance of incorporating culturally diverse datasets and microbiome profiles in future research to ensure broader applicability and equity in AI-driven nutrition systems. (sentence 519-522)

Comment 8: The review lacks a theoretical foundation (e.g., behavior change theories, precision nutrition frameworks) to explain AI’s mechanisms in dietary interventions. For instance, how does AI-driven “personalized feedback” align with psychological models of habit formation? Were socioecological factors integrated?

Response 8: We thank the reviewer for this important observation. We agree that incorporating a theoretical framework would help contextualize the mechanisms by which AI facilitates dietary behavior change and personalization. In our revised Discussion (Section 4.6 line 555-562), we now briefly discuss how AI-driven feedback loops align with behavior change theories—particularly constructs such as reinforcement, self-monitoring, and goal-setting found in models like the Transtheoretical Model and Social Cognitive Theory.

We also acknowledge that most included studies lacked explicit grounding in psychological or behavioral theory. Similarly, few studies integrated socioecological determinants such as food access, cultural preferences, or household dynamics into their algorithmic design, which limits the real-world applicability of AI interventions. We now emphasize this gap and recommend that future AI-based dietary tools be developed with stronger alignment to both precision nutrition frameworks and behavioral science principles.

Reviewer 4 Report

Comments and Suggestions for Authors

The authors make a comprehensive review of the actual efforts to develop personalized dietary recommendations for specific chronical diseases such as diabetes. The considered use-cases from the explored literature include advanced AI-based and Data Science methodologies. My main remark is about the conclusions section that should be clearly separated from the Discussion. This is why I suggest the author to add the 5th section named Conclusions in which they should briefly point out the main achievements from the cited works in the addressed topics. This section could have a few paragraphs but anyway it should be separated from the Discussion section. Usually the scientific articles are ending with concluding sections. The adding of o short Conclusion section should be not very time consuming and it would only require to take some ideas from Discussion and moving into a separate final section.

Author Response

Comment: The authors make a comprehensive review of the actual efforts to develop personalized dietary recommendations for specific chronical diseases such as diabetes. The considered use-cases from the explored literature include advanced AI-based and Data Science methodologies. My main remark is about the conclusions section that should be clearly separated from the Discussion. This is why I suggest the author to add the 5th section named Conclusions in which they should briefly point out the main achievements from the cited works in the addressed topics. This section could have a few paragraphs but anyway it should be separated from the Discussion section. Usually the scientific articles are ending with concluding sections. The adding of o short Conclusion section should be not very time consuming and it would only require to take some ideas from Discussion and moving into a separate final section

Response: We appreciate the reviewer’s suggestion. In response, we have added a new Section 5 (Conclusions) to clearly summarize the key findings of our review. This section highlights the principal contributions, synthesizes the clinical and methodological insights, and reiterates critical limitations and future directions. We believe this improves the structure and clarity of the manuscript in line with standard scientific reporting practices.

Reviewer 5 Report

Comments and Suggestions for Authors

It is a systematic review, it is very descriptive in nature. It is basically putting together the many articles and blending them in one place.  Here are detailed comments:

  1. First the article is a systematic review on studies that compare AI-dietary advice to traditional dietary advice and how that is associated with better clinical outcomes. I thank the authors for the interesting topic.
  2. However, the two major problems I have with this article are; one is a large number of articles were simply excluded by the three authors without us knowing exactly how the exclusions were exactly made, the reasons for excluding these articles must be mentioned. 
  3. The other major problem is that the study does not provide a novel idea. We all know that AI-driven dietary advice would on average outperform traditional advice and we all know that this comes at the token of ethical factors as well as practicality of applying AI. An alternative approach would have been for example showing us where AI helped and where exactly it did not help. Or generating a new piece of knowledge that the average reader cannot agree with except if they look at the analysis and have this one manuscript we are studying now. So yes, I feel the study does not provide that one novel idea or new piece of information which we as AI healthcare average experts need to see.
  I know they are abbreviated but there is not much to analyze in the article. It is a simple collect article and present their findings in this new manuscript so there is not much to analyze or anything like that. I recommend that the authors reanalyze their studies and provide a novel conclusion. For example, if I am diabetic and am more likely to benefit from AI, how, on what aspects, when? And so on.

Author Response

Comment 1: one is a large number of articles were simply excluded by the three authors without us knowing exactly how the exclusions were exactly made, the reasons for excluding these articles must be mentioned. 

Response 1: We thank the reviewer for raising this important point. In the revised manuscript, we have clarified our exclusion process more explicitly in both the Methods section and the PRISMA flow diagram. Specifically, we added detailed exclusion categories for the 1,105 articles removed during title and abstract screening, including: (1) Not AI-generated dietary recommendations, (2)Rule-based intervention, (3)Non-personalized interventions, (4)Non-human studies, (5)Irrelevant outcomes (6)Conference abstracts or protocol-only papers. These categories are now reflected in the updated PRISMA diagram to enhance transparency. Furthermore, we explicitly list the reasons for full-text exclusions (e.g., conference abstracts, AI-generated supplements, ongoing trials) in the Results section (Section 3.1).

Comment 2: The other major problem is that the study does not provide a novel idea. We all know that AI-driven dietary advice would on average outperform traditional advice and we all know that this comes at the token of ethical factors as well as practicality of applying AI. An alternative approach would have been for example showing us where AI helped and where exactly it did not help. Or generating a new piece of knowledge that the average reader cannot agree with except if they look at the analysis and have this one manuscript we are studying now. So yes, I feel the study does not provide that one novel idea or new piece of information which we as AI healthcare average experts need to see.

  I know they are abbreviated but there is not much to analyze in the article. It is a simple collect article and present their findings in this new manuscript so there is not much to analyze or anything like that. I recommend that the authors reanalyze their studies and provide a novel conclusion. For example, if I am diabetic and am more likely to benefit from AI, how, on what aspects, when? And so on.

Response 2: We appreciate the reviewer’s concern regarding the perceived lack of novelty. In response, we have revised the manuscript to more clearly articulate the unique contributions of our review.

Specifically, this review offers a structured comparative synthesis of how AI-generated dietary interventions have been applied across diverse clinical populations, intervention designs, and health outcomes. Rather than simply summarizing existing studies, we now emphasize:

  1. Heterogeneity in AI efficacy: In Section 4.6 (Critical Gaps and Future Directions), we highlight how AI-generated diets yielded varying degrees of improvement depending on the health domain (e.g., glycemic control vs. cholesterol outcomes) and population context (e.g., IBS vs. prediabetic patients). For instance, while glycemic markers improved significantly in Ben-Yacov et al., no comparable benefits were seen in cholesterol-HDL ratio—suggesting outcome-specific limitations of personalization.
  2. Clinical uncertainty and domain-specific utility: We now explicitly point out where AI interventions underperformed or showed marginal improvements, drawing attention to outcome-level inconsistencies (e.g., limited sustainability or no long-term superiority in some studies).
  3. Framework for AI effectiveness: We propose a conceptual typology (see new Section 4.7) outlining when AI is more likely to be beneficial, based on factors such as input data richness, feedback frequency, population characteristics, and health outcome types. This provides actionable insights for future trials and real-world implementation.
  4. Ethical-practical synthesis: We integrated a more detailed discussion (in Section 4.6) on how ethical challenges (e.g., algorithmic opacity, data privacy) intersect with practical limitations such as demographic bias and lack of cost-effectiveness assessments—emphasizing that the same mechanisms enabling personalization may also introduce harm if unchecked.

Together, the revisions made in response to this comment aim to move beyond mere aggregation by offering transferable insights for both practitioners and researchers. We hope these additions better articulate the unique contributions and novel value our review brings to the existing literature.

Round 2

Reviewer 3 Report

Comments and Suggestions for Authors

I believe the author has adequately addressed my concerns about this article and recommend its publication.

Reviewer 5 Report

Comments and Suggestions for Authors

Improved the novelty in the revision. Thank you. This is a very interesting review. now.